# The Microbiome Structure of a Rice-Crayfish Integrated Breeding Model and Its Association with Crayfish Growth and Water Quality

Ling Chen,[a] Jin Xu,[b] Weitao Wan,[a] Zhiwei Xu,[a] Ruixue Hu,[a] Yunzeng Zhang,[c] Jinshui Zheng,[d,e] Zemao Gu[a,f]

aCollege of Fisheries/Shuangshui Shuanglü Institute, Huazhong Agricultural University, Wuhan, China

bCitrus Research and Education Center, Department of Microbiology and Cell Science, IFAS, University of Florida, Lake Alfred, Florida, USA

cJoint International Research Laboratory of Agriculture and Agri-Product Safety, the Ministry of Education of China, Yangzhou University, Yangzhou, China

dState Key Laboratory of Agricultural Microbiology, Huazhong Agricultural University, Wuhan, China

eHubei Key Laboratory of Agricultural Bioinformatics, Huazhong Agricultural University, Wuhan, China

fHubei Hongshan Laboratory, Wuhan, China

Ling Chen and Jin Xu contributed equally to this article. Author order was determined alphabetically.

**ABSTRACT** The rice-crayfish (RC) integrated breeding model is an important and special agricultural ecosystem that provides a unique ecological environment for exploring the microbial diversity, composition, and functional capacity. To date, little is known about the effect of the breeding model on microbiome assembly and breeding model-specific microbiome composition and the association of the microbiome with water quality and crayfish growth. In the present study, we assessed the taxonomic shifts in gut and water microbiomes and their associations with water quality and crayfish growth in the RC and crayfish monoculture (CM) breeding models across six time points of rice growth, including seedling (a), tillering and jointing (b), blooming (c), filling (d), fruiting (e), and rotting of rice residues (f). Dominant bacterial phyla, such as *Proteobacteria*, *Actinobacteria*, *Bacteroidetes*, and *Firmicutes*, were detected in both gut and water microbiomes across breeding models. Notably, the diversity and structure of the gut and water microbiomes significantly ($P <$ 0.001) differed between the RC and CM models, with higher microbial diversity being noted in the RC model than in the CM model. The taxa enriched in the RC model included *Bacillus* sp., *Streptomyces* sp., *Lactobacillus* sp., *Prevotella* sp., *Rhodobacter* sp., *Bifidobacterium* sp., *Akkermansia* sp., and *Lactococcus* sp., some of which are potentially beneficial to animals. However, opportunistic pathogens, such as *Citrobacter* sp. and *Aeromonas* sp., were depleted in the RC model. Furthermore, in the RC model, the enriched taxa that formed complex cooccurrence networks showed a significant positive correlation with water quality and crayfish growth, whereas the depleted taxa showed a significant negative correlation with water quality and crayfish growth. These results suggest that the RC model has a better microbiome composition and that RC model-specific microbes could play important roles in improving crayfish growth and water quality.

**IMPORTANCE** The present study comprehensively compared two different breeding models in terms of their microbiome composition and the associations of the microbiomes with crayfish growth and water quality. RC model-specific microbiome composition was identified; these microbes were found to have a positive association with water quality and crayfish growth. These results provide valuable information for guiding microbial isolation and culture and for potentially harnessing the power of the microbiome to improve crayfish production and health and water quality.

Address correspondence to Zemao Gu, guzemao@mail.hzau.edu.cn.

The authors declare no conflict of interest.

**KEYWORDS** cooccurrence networks, correlationship, crayfish growth, functional prediction, high-throughput sequence, keystone, microbiome structure, rice-crayfish integrated breeding model, water quality

Rice-aquatic animal (RAA) integrated breeding models, which combine agriculture and aquaculture, have been practiced in Asian areas for over 2,000 years (1). Compared with aquatic animal monoculture models, RAA models can comprehensively utilize water, land, biological resources, and nutritional resources, with reduced usage of fertilizers and pesticides (2). Previous studies and diverse practices have suggested that RAA models can improve the environmental quality of the ecosystem (3), in addition to improving the biodiversity (4) and yield (5, 6). The rice-crayfish (RC) integrated breeding model is a type of RAA model that originated in Louisiana, USA, several decades ago (7). It has become a primary cultivation model for crayfish in the waterlogged areas of China (8). According to the Crayfish Industry Report 2020 (http://www.moa.gov.cn/), the RC model has been practiced in China on a 1.1-million-ha area and the production of crayfish has been more than 1.78 million tons, accounting for 47.71% of the total RAA model area and 85.96% of the total crayfish culture, respectively.

There are complex interactions among the crayfish, plants, microbiome, and environment in the RC model. Of these, the microbiome plays indispensable roles in nutrient cycling, biogeochemical processes, and the maintenance of host growth and health, among others (9–11). For example, aquatic microbes play important roles in matter and energy recycling, and some of these microbes may contribute to the reduction of eutrophication pollution (12). The gut microbiota of animals is closely related to their health status (13–15), nutrient metabolism (14), and immune system functioning (15). Completely understanding the microbiome composition, microbiome function, and roles of the microbiome in host growth and environmental quality in the RC model could help improve productivity and environmental quality during crayfish cultivation. So far, only limited studies have focused on the microbiome in the RC model. The 16S rRNA high-throughput sequencing approach has been used to assess the composition and diversity of bacterial communities in crayfish gut, water, or sediment in the RC model (16–20). Two previous studies have suggested that there was no significant difference in gut, water, and sediment microbiomes between the RC and crayfish monoculture (CM) models (18, 19). Zhang et al. suggested that overutilizing the RC model can lead to the destruction of soil microbial ecology (20). However, the effect of the breeding model on microbiome assembly and composition and the association of the microbiome with water quality and crayfish growth remain largely unknown.

In the present study, we hypothesized that the breeding model is one of the most important driving factors for microbiome assembly and that breeding model-specific microbes are closely related to water quality and crayfish growth. Therefore, we used 16S rRNA gene high-throughput sequencing to comprehensively assess the response of the microbiome of crayfish gut and associated water to breeding models. We collected crayfish and water samples between two breeding models across six time points of rice growth, including seedling (a), tillering and jointing (b), blooming (c), filling (d), fruiting (e), and rotting of rice residues (f) (Fig. 1; see also Fig. S1 in the supplemental material). We aimed to explore the following: (i) the diversity and structure of water and crayfish gut microbiomes in two breeding models, (ii) the impact of the breeding model on the water and gut microbiome assembly and functional capacity, and (iii) the RC-specific microbes and their associations with water quality and crayfish growth.

## RESULTS

**Overview of gut and water microbiomes in the RC and CM models.** Through 16S rRNA gene amplicon sequencing, we conducted microbial community profiling using 29 and 28 water samples from the RC and CM models, respectively, and 28 and 25 crayfish gut samples from the RC and CM models, respectively (see Table S1 in the supplemental material). We obtained a total of 4,498,664 high-quality sequences of the 16S

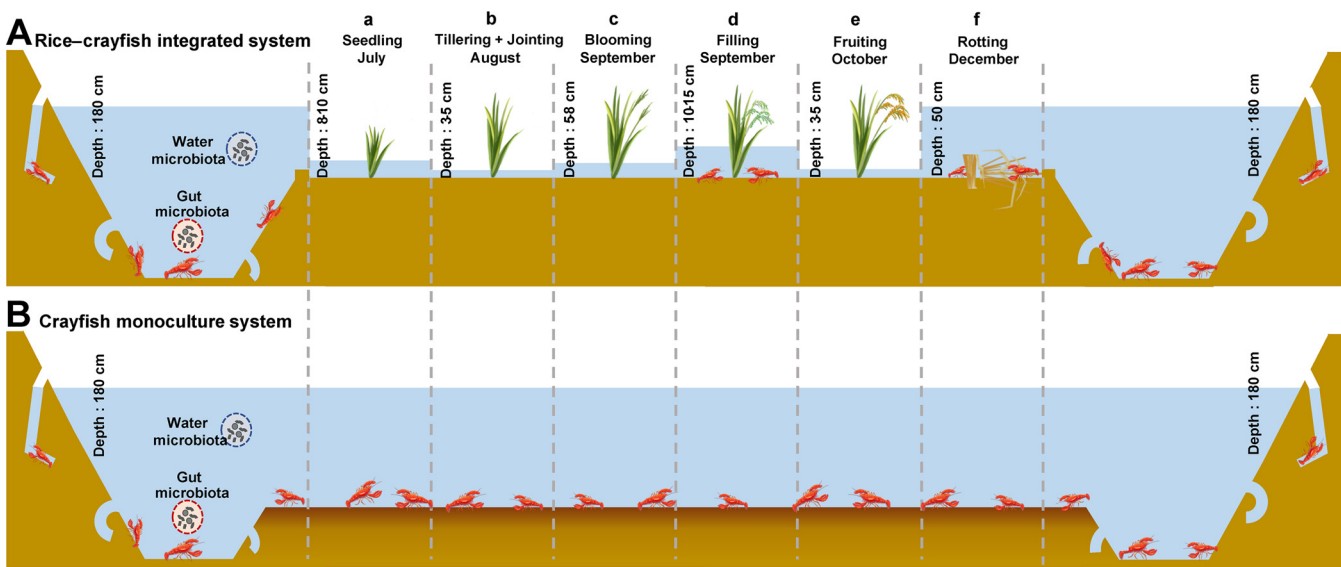

**FIG 1** Two different breeding models for crayfish. (A) The rice-crayfish integrated breeding model (RC). (B) The crayfish monoculture breeding model (CM). The RC model combined agriculture and aquaculture in the paddy field. In this model, mid-rice was grown with crayfish during June to October generally. There are six developmental stages of rice, which are as follows: a, seedling of rice (generally last 2 weeks after transplant to fields from June to July); b, tillering and jointing of rice (from July to August); c, blooming of rice (September); d, filling of rice (from September to October); e, fruiting of rice (October); and f, rotting of rice residues (the rice residues were returned into paddy fields after harvest until the next year). During this period, crayfish mainly lived in the ditch that surrounded the paddy field. The depth of water was regulated to satisfy the needs of rice. In the CM model, only crayfish were cultured all year and the water depth was kept stable.

rRNA gene V3-V4 region (range 20,309 to 59,191, Table S1). These sequences were clustered into 4,819 operational taxonomic units (OTUs), which were assigned to 46 bacterial phyla (Table S2). Rarefaction analysis of the OTU number revealed that the detected OTUs for both gut and water microbiomes reached a plateau, suggesting that we had captured most of the observed OTU richness (Fig. S2). In the water samples, the dominant bacterial phyla identified between two breeding models included *Proteobacteria*, *Actinobacteria*, *Bacteroidetes*, *Firmicutes*, *Cyanobacteria*, and *Verrucomicrobia* (Fig. 2A). In the gut samples, the dominant bacterial phyla identified between two breeding models across multiple time points included *Firmicutes*, *Bacteroidetes*, *Tenericutes*, *Proteobacteria*, RsaHF231, and *Actinobacteria* (Fig. 2B).

**Differences in microbiome structure and function between the RC and CM models.** The alpha diversity of water and gut microbiomes significantly differed between the RC and CM models across multiple time points. Both water and gut microbiomes showed significantly higher diversity in the RC model than in the CM model ($P < 0.001$, Fig. 3A and B). Nonmetric multidimensional scaling (NMDS) with analysis of similarities (ANOSIM) also revealed that the structure of water and gut microbiomes significantly differed between the RC and CM models across multiple time points ($P < 0.001$, Fig. 3C and D). In this study, the type of breeding model contributed to a higher variation in the gut microbiome ($R = 0.423$) than in the water microbiome ($R = 0.200$) (Fig. 3C and D). Notably, the water samples from time point f (rotting of rice residues) were very different from others in the RC model (Fig. 3C).

To identify the major differential taxa of water and gut microbiomes between two breeding models, we compared the relative abundances of microbes in water and gut samples between the RC and CM models at the phylum and OTU levels (corrected $P < 0.05$, Fig. 4; see also Fig. S3 and Tables S3 to S6). In the water samples, the abundances of *Bacteroidetes* and *Verrucomicrobia* were significantly lower in the RC model than in the CM model (corrected $P < 0.05$, Table S3). At the OTU level, 239 OTUs of the water microbiome were enriched in the RC model. On the other hand, 56 OTUs were depleted in the RC model. The RC-enriched OTUs mainly belonged to the phyla *Proteobacteria*, *Actinobacteria*, *Bacteroidetes*, *Cyanobacteria*, and *Chloroflexi*, while the RC-depleted OTUs mainly belonged to *Proteobacteria* and *Bacteroidetes* (Table S5). In

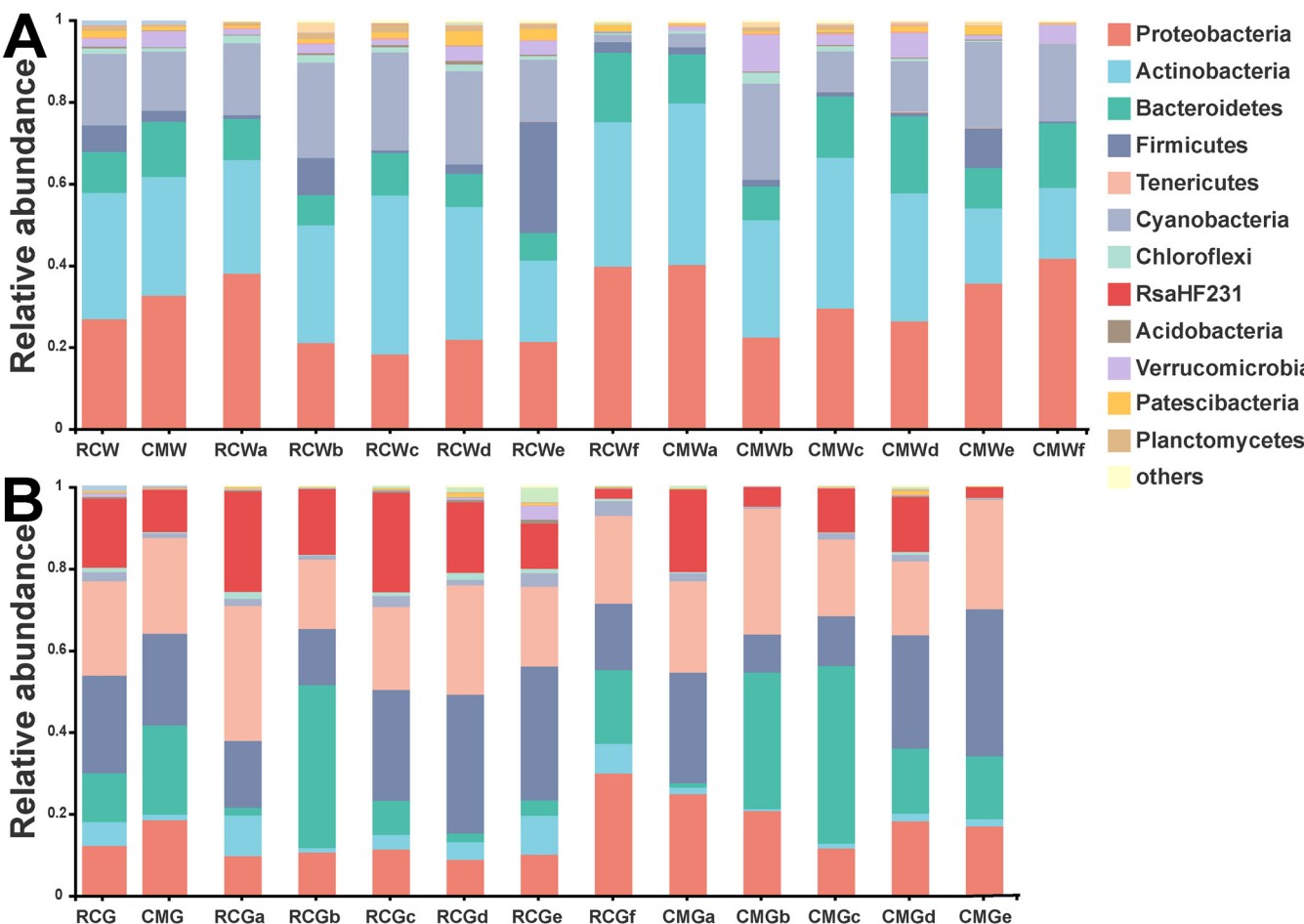

**FIG 2** Taxonomic comparisons of gut and water microbiomes between the RC and CM models across multiple time points. (A) Phylum-level comparisons of water microbiomes between the RC and CM models across multiple time points. (B) Phylum-level comparisons of gut microbiomes between the RC and CM models across multiple time points.

the gut samples, the abundance of *Actinobacteria* was significantly higher while the abundances of *Firmicutes*, *Bacteroidetes*, *Tenericutes*, and *Proteobacteria* were significantly lower in the RC model than in the CM model (corrected $P < 0.05$, Table S4). At the OTU level, 59 OTUs of the gut microbiome were enriched, while 31 OTUs were depleted in the RC model. The RC-enriched OTUs mainly belonged to the phyla *Proteobacteria*, *Actinobacteria*, *Firmicutes*, and *Chloroflexi*, while the RC-depleted OTUs mainly belonged to *Proteobacteria* and *Firmicutes* (Fig. 4; see also Fig. S3 and Table S6). Remarkably, several RC-enriched OTUs of the water or gut microbiome were affiliated with microbes potentially beneficial to aquaculture (21), such as *Bacillus* sp., *Streptomyces* sp., *Lactobacillus* sp., and *Lactococcus* sp., while several RC-depleted OTUs were affiliated with opportunistic crayfish pathogens (22), such as *Citrobacter* sp. and *Aeromonas* sp. Furthermore, we constructed cooccurrence networks of both gut and water microbiomes to explore the relationship among the RC-enriched/depleted OTUs. The results indicated that the RC-enriched OTUs formed a much more complex network with strong Spearman's correlations than the RC-depleted OTUs (Fig. S4). One OTU of the gut microbiome assigned to *Streptomyces* sp. positively correlated with other OTUs assigned to genera belonging to *Actinobacteria*, such as *Conexibacter* sp. One OTU of the gut microbiome assigned to *Citrobacter* sp. negatively correlated with OTUs assigned to *Sphingomonas* sp. and uncultured *Chthoniobacterales* LD29 (Fig. S4 and S5).

Furthermore, we identified the indicator OTUs and keystone OTUs of water and gut microbiomes in the RC and CM models (Fig. 5; see also Tables S7, S8, and S9). In

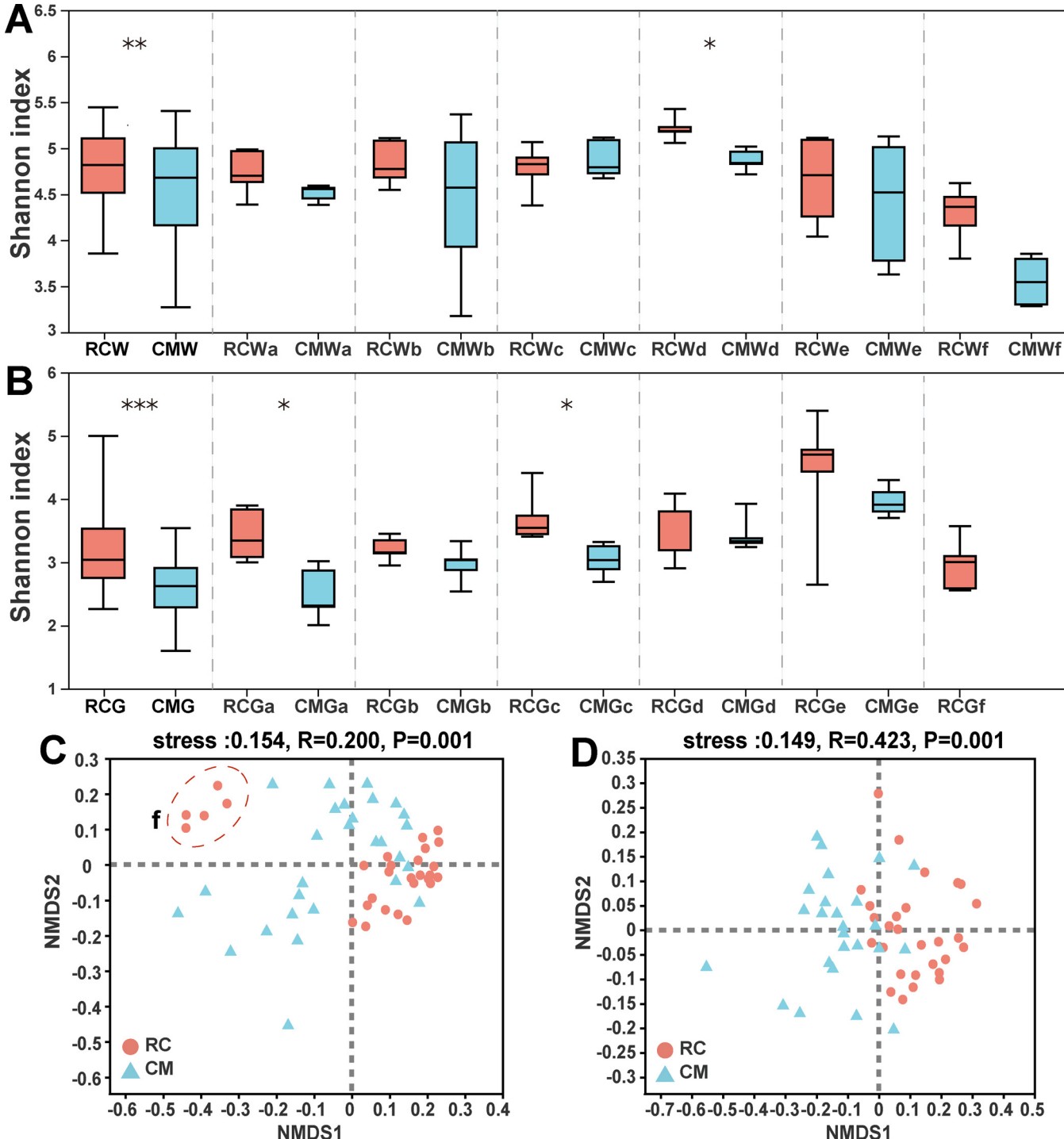

**FIG 3** Diversity comparisons of gut and water microbiomes between the RC and CM models across multiple time points. (A) Alpha diversity comparisons of water microbiomes between the RC and CM models across multiple time points; *, $P < 0.05$; **, $P < 0.01$; ***, $P < 0.001$; paired Wilcoxon rank sum test; center value represents the median of Shannon index. (B) Alpha diversity comparisons of gut microbiome between the RC and CM models across multiple time points; *, $P < 0.05$; **, $P < 0.01$; ***, $P < 0.001$; paired Wilcoxon rank sum test; center value represents the median of Shannon index. (C) Beta diversity comparisons of water microbiome between the RC and CM models. (D) Beta diversity comparisons of gut microbiomes between the RC and CM models.

addition, keystone taxa of water and crayfish gut microbiomes were identified as well (Table S9). In the water microbiome, 26 and 34 indicator OTUs were identified for the RC and CM models, respectively. The RC-indicator OTUs of the water microbiome mainly belonged to *Proteobacteria* and *Firmicutes*, while CM-indicator OTUs of the water microbiome mainly belonged to *Proteobacteria* and *Bacteroidetes* (Fig. 5A; see

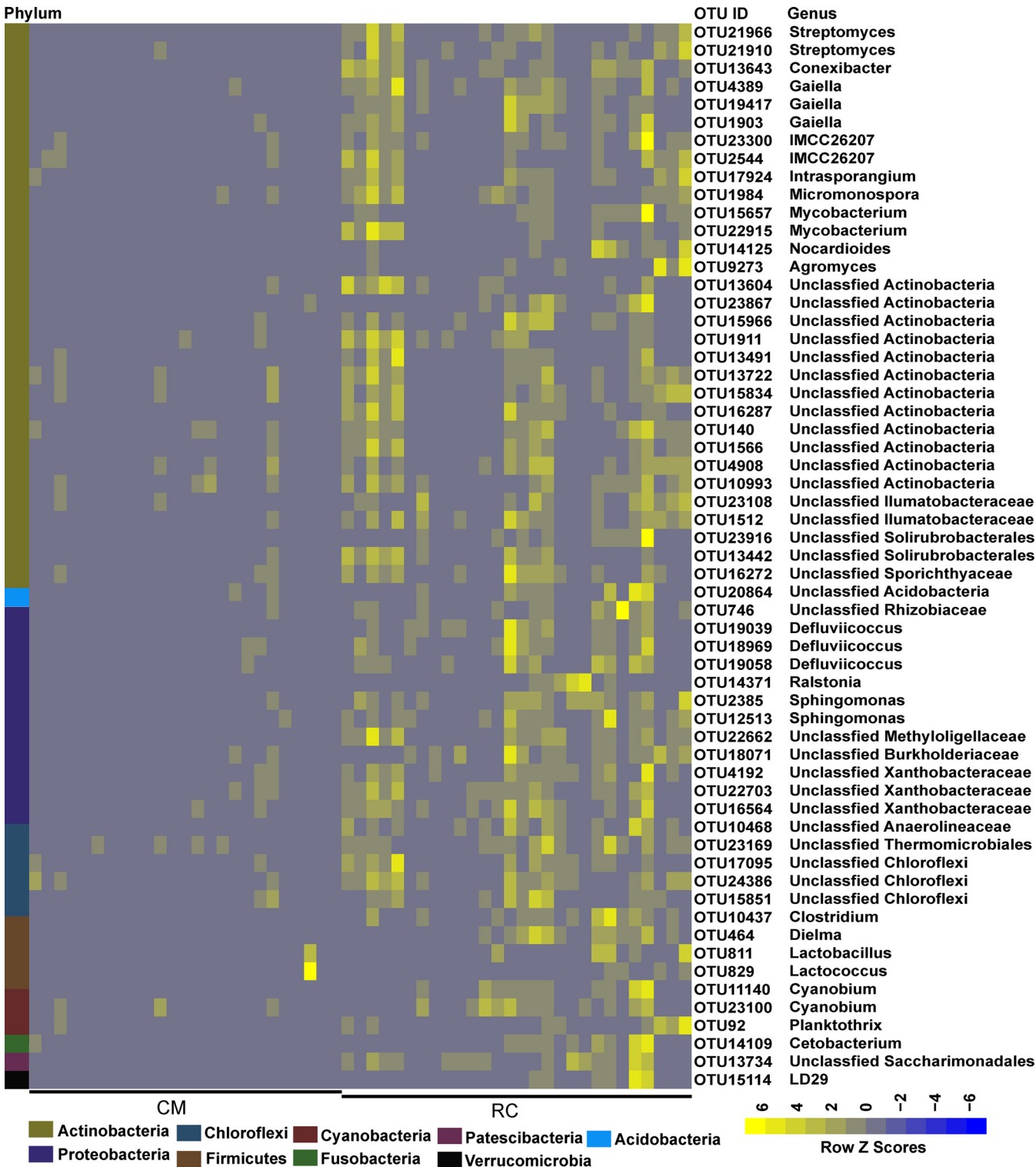

**FIG 4** Relative abundances of the RC-enriched gut microbiomes at OTU level. Scale, the relative abundance of OTU at row normalization by removing the mean (centering) and dividing by the standard deviation (scaling). The color from blue to yellow represents a relative abundance of each taxon from low to high.

also Table S7). In the gut microbiome, 45 and 6 indicator OTUs were identified for the RC and CM models, respectively. The RC-indicator OTUs of the gut microbiome mainly belonged to *Proteobacteria*, *Actinobacteria*, and *Chloroflexi*, while CM-indicator OTUs of the gut microbiome mainly belonged to *Firmicutes* (Fig. 5B and also Table S8).

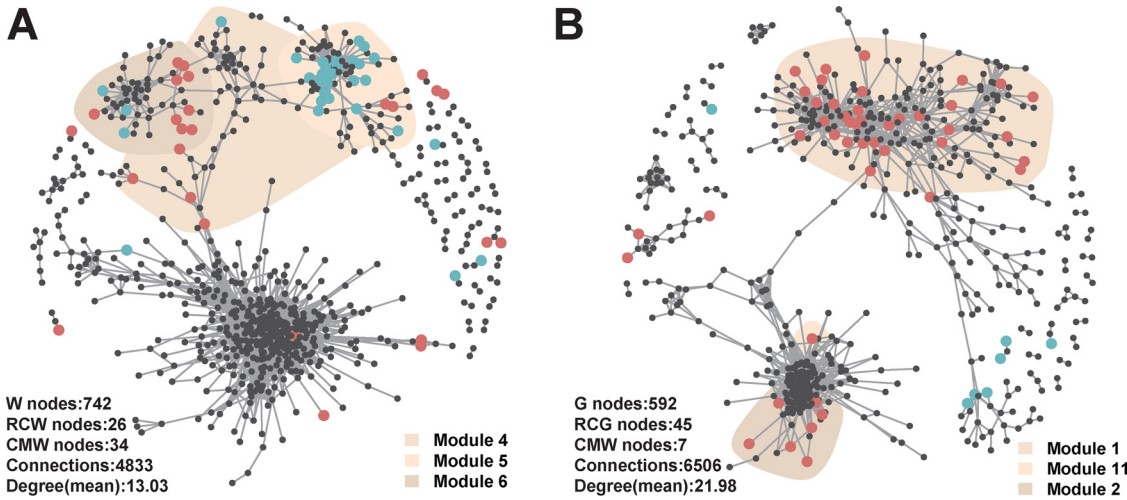

**FIG 5** Cooccurrence patterns of indicator OTUs for water (A) and gut (B) microbiomes. Cooccurrence networks visualizing significant correlations ($\rho > 0.7$, $P < 0.001$; indicated with gray lines) between OTUs. OTUs are colored by their association with the different breeding models (red and blue OTUs are indicator OTUs for the RC and CM models, respectively; gray OTUs are nonindicator OTUs for breeding models).

Particularly, we found that several RC-indicator OTUs of the gut microbiome were affiliated with bacteria potentially beneficial to the animals (21), such as *Bacillus* sp., *Streptomyces* sp., *Prevotella* sp., *Rhodobacter* sp., *Bifidobacterium* sp., and *Akkermansia* sp. At the same time, several CM-indicator OTUs were affiliated with opportunistic crayfish pathogens (22), such as *Citrobacter* sp. and *Aeromonas* sp. Through cooccurrence network analysis, 6 and 23 keystone OTUs were identified from the water and gut microbiomes, respectively. The keystone OTUs of both water and gut microbiomes did not overlap the indicator OTUs and showed higher abundances in the RC model than in the CM model. The keystone OTUs of the water microbiome mainly belonged to *Gammaproteobacteria*, while those of the gut microbiome mainly belonged to several classes, including *Bacilli*, *Clostridia*, and *Bacteroidia* (Table S9).

Across breeding models, the predicted KEGG orthologs (KOs) of gut and water microbiomes were mainly involved in 24 KEGG level 2 pathways. Some pathways of the gut microbiome significantly differed in abundance between the RC and CM models. For example, the abundance of pathways involved in terpenoid and polyketide metabolism in the gut microbiome was significantly higher in the RC model than in the CM model (corrected $P < 0.05$, Fig. 6). In both water and gut microbiomes, the abundances of antibiotic synthesis-related pathways, such as those related to the synthesis of enediyne antibiotics, tetracycline, prodigiosin, staurosporine, and ansamycins, were significantly higher in the RC model than in the CM model. Furthermore, the abundances of antimicrobial resistance-related pathways, such as those related to cationic antimicrobial peptide and beta-lactam resistance, were significantly lower in the RC model than in the CM model (corrected $P < 0.05$, Fig. S6).

**Correlation of microbes with water quality and crayfish growth.** In this study, the water physicochemical data significantly differed at different degrees between the RC and CM models across six time points of rice growth, including seedling (a), tillering and jointing (b), blooming (c), filling (d), fruiting (e), and rotting of rice residues (f) (the details of water physicochemical data are given in Table S10). For example, compared to the CM model, the concentrations of total nitrogen (TN), total phosphorus (TP), ammonia ($NH_4^+$-N), and nitrite ($NO_2^-$-N) in the RC model were significantly higher at time point a (seedling of rice) but lower at other time points (b, c, d, e, and f) ($P < 0.05$, Fig. S7). Similarly, the crayfish growth data significantly differed at different degrees between the RC and CM models across six time points of rice growth (the details of growth data are given in Table S11). For example, the total length (TL) and total weight

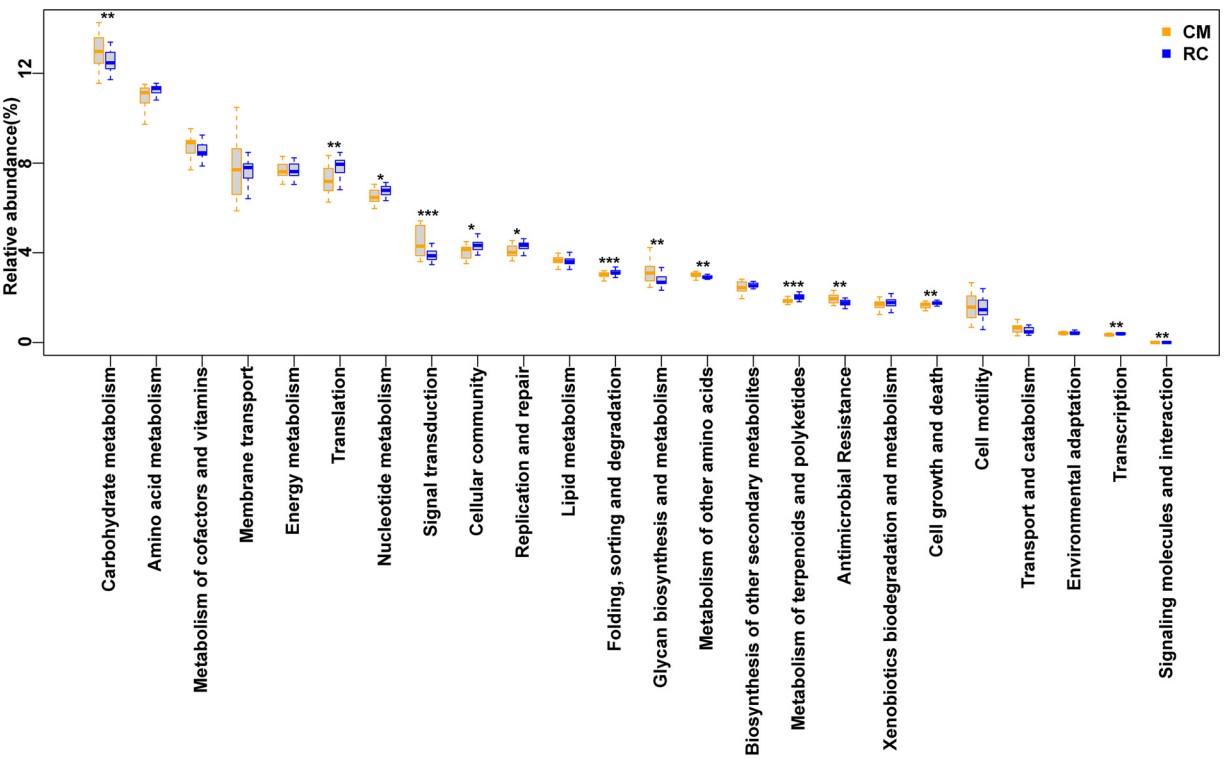

**FIG 6** Comparison of the functional KEGG level 2 pathways of gut microbiomes between the RC and CM models. The center value represents the median of the relative abundance. *, $P < 0.05$; **, $P < 0.01$; ***, $P < 0.001$. $P$ value was generated using paired DESeq2.

(TW) of crayfish were significantly higher in the RC model than in the CM model at all time points, except for time point a (seedling of rice) ($P < 0.05$, Fig. 7).

Next, we assessed the associations of water physicochemical data and crayfish growth data with their respective microbiomes. Both types of data were significantly correlated with their respective microbiomes, suggesting that these factors affected the water and gut microbiome assembly ($P < 0.05$, Tables S12 and S13). Furthermore, we assessed the associations of water physicochemical data and of crayfish growth data with RC-enriched/depleted OTUs. In the water microbiome, RC-enriched OTUs showed a significant negative correlation with the TN concentration, TP concentration, and pH, while most RC-depleted OTUs showed a significant positive correlation with the TN concentration and negative correlation with the TP concentration. Specifically, RC-enriched OTUs belonging to *Proteobacteria*, *Cyanobacteria*, *Bacteroidetes*, and *Actinobacteria* showed a significant negative correlation with the TN concentration, while RC-depleted OTUs belonging to *Bacteroidetes*, *Proteobacteria*, and *Verrucomicrobia* showed a significant positive correlation with the TN concentration ($P < 0.05$, Fig. S8). In the gut microbiome, most RC-enriched OTUs showed a significant positive correlation with crayfish growth, while most RC-depleted OTUs showed a significant negative correlation with crayfish growth. Specifically, RC-enriched OTUs belonging to *Streptomyces* sp., *Conexibacter* sp., *Clostridium* sp., and *Lactococcus* sp. showed a significant positive correlation with crayfish growth, while the RC-depleted OTU belonging to *Citrobacter* sp. showed a significant negative correlation with crayfish growth ($P < 0.05$, Fig. 8).

In addition, indicator and keystone OTUs of the water microbiome showed a significant association with the water physicochemical data, and those of the gut microbiome showed a significant association with the crayfish growth data ($P < 0.05$, Fig. S9). For example, keystone OTUs of the water microbiome that had higher abundances in the RC model than in the CM model showed a significant negative correlation with the TN concentration, TP concentration, and pH. Indicator OTUs of the water microbiome showed a significant negative correlation with the $NO_2^-$-N concentration in the

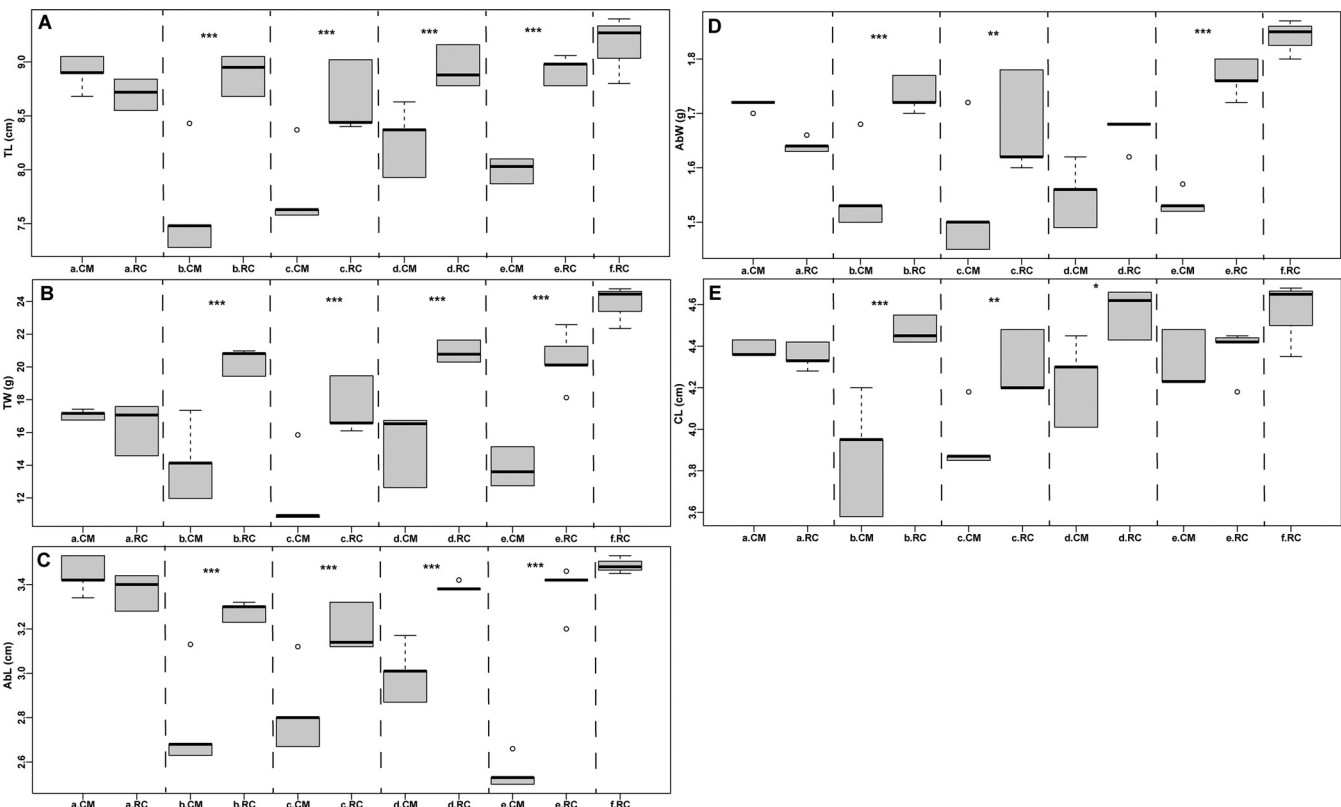

**FIG 7** Phenotype comparisons of crayfish between the RC and CM models. (A) Total length (TL); (B) total weight (TW); (C) abdomen length (AbL); (D) abdomen weight (AbW); (E) carapace length (CL). *, $P < 0.05$; **, $P < 0.01$; ***, $P < 0.001$. $P$ value was generated using paired ANOVA and Tukey HSD method; center value represents the median of data for each phenotype.

RC model but a significant positive correlation with the $NO_2^-$-N concentration in the CM model (Fig. S9A). Moreover, RC-indicator OTUs of the gut microbiome showed a significant positive correlation with crayfish growth. Specifically, RC-indicator OTUs belonging to *Streptomyces* sp., *Prevotella* sp., and *Bifidobacterium* sp. showed a significant positive correlation with crayfish growth (Fig. S9B).

## DISCUSSION

In the present study, we found that several bacterial phyla, such as *Proteobacteria*, *Actinobacteria*, *Bacteroidetes*, and *Firmicutes*, were dominant in both gut and water microbiomes of two breeding models. These findings are similar to previous findings in microbiomes of rearing water and guts of other crustaceans (16–20). Compared to the CM model, the diversity and community structure of both gut and water microbiomes were significantly different in the RC model, suggesting that the breeding model may affect the microbiome assembly of crayfish gut and water. However, these findings were not consistent with the previous study, in which water, sediment, and crayfish samples from a single time point were collected, and no statistically significant differences of microbiome diversity and structure between the RC and CM models were found (19). Due to the different environment status of the RC model at different times, we supposed that the sampling time could be a reasonable factor causing the different results. Numbers of studies have reported a variable microbiome structure at different sampling times (21–23). Also, our results showed a fluctuating diversity and structure of both water and gut microbiomes across six sampling times. Therefore, we used a multiple-time-point sampling strategy to comprehensively capture the microbiome for the RC and CM models.

Many factors may contribute to the microbial differences between these two breeding models. Nutrients are one of the most important factors that shape microbiome

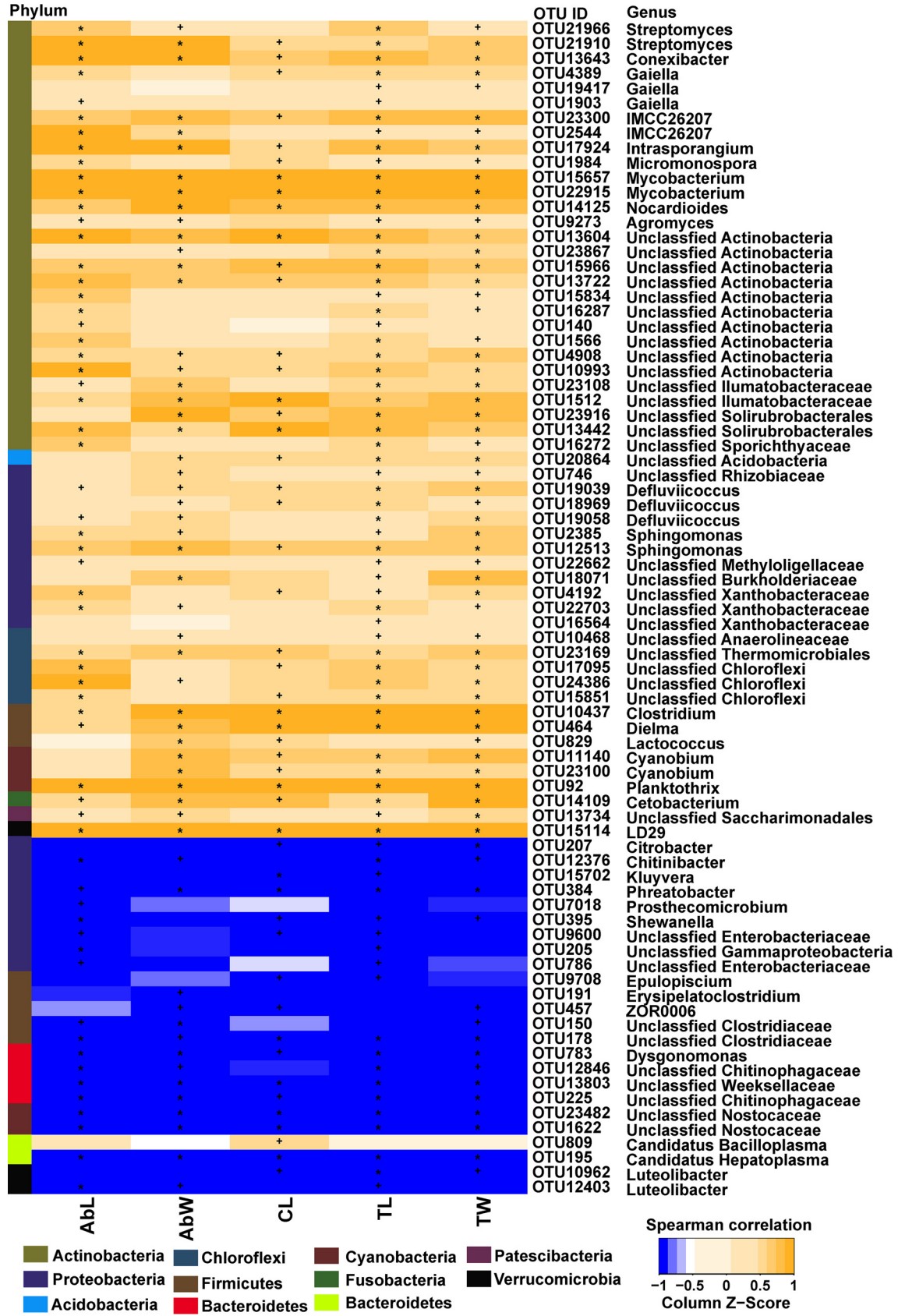

FIG 8 The correlation between phenotypes and gut microbiome. Orange denotes positive Spearman correlation, while blue denotes negative Spearman correlation. An asterisk denotes $P < 0.01$; a plus sign denotes $P < 0.05$. TL, total length; TW, total weight; AbL, abdomen length; AbW, abdomen weight; CL, carapace length.

structure (24). More plant-derived nutrients for microbes, such as fibers, have been reported in the RC model than in the CM model (23, 24). Fiber is a good source of microbiota-accessible carbohydrates, which can be utilized by microbes and can increase microbial diversity (25). Thus, the higher diversity of both water and gut microbiomes in the RC model may partially be explained by the higher fiber content. Consistent with this finding, the structure of both gut and water microbiomes in the RC model at time point f significantly differed from that at other time points because of the organic fertilizer from the rice residues (24). Some members of *Actinobacteria* that contribute significantly to the turnover of complex biopolymers and decomposition of organic matter (25, 26) had higher abundance in the RC model than in the CM model. Moreover, gut microbes positively correlated with fiber intake (27), such as *Ruminococcaceae* sp., *Bifidobacteria* sp., and lactic acid bacteria (28), were more abundant in the RC model than in the CM model. Consistent with the finding that plant-derived nutrients contain diverse terpenoids (29), the functional traits of gut microbes involved in terpenoid metabolism were found to be enriched in the RC model. In the RC model, more nitrogen sources are required for rice growth. Besides artificial nitrogen sources, microbes can help plants to generate nitrogen sources from the environment (30). Accordingly, some members of *Cyanobacteria* involved in nitrogen fixation (31) were found to be enriched in the water microbiome in the RC model. In addition, some members of *Chloroflexi* involved in nitrification (32) and *Bacillus* sp. involved in ammonification (33) had a higher abundance in the water microbiome in the RC model than in the CM model. Sediment that contained organic matter and nitrogen has been suggested as another important factor that may affect microbiome composition (34, 35). In the future, we need to collect the physicochemical factors and microbiome of sediment across multiple times to further study the effect of sediment on the microbiome assembly of the RC model.

As a typical rice-aquatic animal breeding model, the RC model can increase the environmental quality of the ecosystem (36), in addition to improving the biodiversity (37) and productivity (38). In the present study, we found that crayfish growth and water quality were better in the RC model than in the CM model. The microbiome is considered to play very important roles in sustainable aquatic breeding models, e.g., nutrient transformation and the promotion of animal growth and health (9–11). The diversity in both water and gut microbiomes was higher in the RC model than in the CM model. In addition, more complex species interactions of water and gut microbiomes were found in the RC model. RC-enriched microbes showed a significant correlation with each other and formed a much more complex network than RC-depleted microbes. According to the insurance hypothesis, biodiversity insures ecosystems against declines in their functioning because more species provide greater guarantees that some will maintain functioning even if others fail (39). Species interactions may affect community structure and drive the stability in microbial ecology (40). Thus, these results indicate that the RC model had a more stable microbiome structure, which could enhance biodiversity and stability of the agroecosystem (41–43). Several microbes potentially beneficial to animals (44), such as *Bacillus* sp., *Streptomyces* sp., *Lactobacillus* sp., *Prevotella* sp., *Rhodobacter* sp., *Bifidobacterium* sp., *Akkermansia* sp., and *Lactococcus* sp., were enriched in the RC model and positively correlated with other enriched microbes, such as those belonging to *Actinobacteria*. In contrast, several opportunistic crayfish pathogens (45), such as *Citrobacter* sp. and *Aeromonas* sp., were depleted in the RC model and negatively correlated with other enriched microbes. Importantly, microbes enriched in the RC model, including those potentially beneficial to animals, showed a significant positive correlation with crayfish growth, suggesting that these microbes can promote the growth and health of crayfish. At the same time, microbes depleted in the RC model, including opportunistic pathogens, showed a significant negative association with crayfish growth, further suggesting that the RC model shaped a much more stable and healthier microbial community.

In aquatic animal monoculture systems, water quality eventually decreases. For example, the concentrations of ammonia, nitrite, and phosphorus increase and the oxygen level decreases, becoming toxic to aquatic animals (46, 47). Although artificial nitrogen and phosphorus fertilizers were added for rice growth in the RC model, we found that the ammonia, nitrite, and phosphorus concentrations were lower in the water samples of the RC model than in those of the CM model, suggesting that the water quality in the RC model was better. Microbes may play important roles in the transformation of nitrogen and phosphorus, which were used for rice growth in the RC model (48). Consistent with this, we found that members of *Cyanobacteria* involved in nitrogen fixation and oxygen synthesis and those of *Chloroflexi* involved in nitrification were enriched in the RC model. Furthermore, in the RC model, enriched microbes in water samples showed a significant negative correlation with the TN and TP concentrations, while most depleted microbes showed a significant positive correlation with the TN concentration. In addition, *Streptomyces* sp. (49) and *Bacillus* sp. (33) can modulate a wide range of water quality parameters in aquaculture systems, including transparency, total dissolved solids, pH, conductivity, chemical oxygen demand, dissolved oxygen, biological oxygen demand, alkalinity, phosphates, nitrogenous species, and hardness, and these were enriched in the RC model. Antibiotics have been overused and have caused antibiotic resistance in aquatic animal monoculture systems (50). Compared with the CM model, antibiotic synthesis-related genes were enriched and antibiotic resistance-related genes were depleted in the RC model, implying that the RC model bears a lower burden of antibiotic resistance. These results suggest that the RC model can shape a better microbial ecosystem than the CM model and that RC model-specific microbes can play important roles in improving crayfish growth and water quality. In the future, RC model-specific microbes can be used as targets for interventional studies in order to assess their role in crayfish productivity and health and in maintaining water quality.

In summary, we identified breeding model-specific microbiome composition and its association with crayfish growth and water quality. The identification of RC model-specific gut and water microbiomes and their strong association with crayfish growth and water quality suggest that these microbes could be used as targets for future microbiome engineering. However, we only unraveled the identity of these microbes and their association with crayfish growth and water quality in two crayfish breeding models; this information is not sufficient to fully understand the microbial function and mechanism for microbiome engineering. In future studies, we need to investigate the potential of these microbes and the roles they play and to identify the causal relationship among the microbiome, host, and environment using multi-omics approaches, including metagenomic, metatranscriptomic, metaproteomic, and metametabolomic analyses and the study of culturable members of the community. In the long run, synthetic communities, as well as genome editing of specific host genes, can be used to address critical issues in sustainable aquatic culture systems, e.g., for protecting hosts against pathogens or biofertilization.

## MATERIALS AND METHODS

**Experimental design and model management.** The experiment was conducted on the up-to-5-year test ground in a waterlogged area during the periods from June to December of 2019 in Yangxin County, Hubei Province (114°58'E, 29°47'N). This region has a subtropical monsoon climate, with an average annual rainfall of 1,389.6 mm. To ensure consistency in the experimental conditions, we selected three representative standardized RC and CM ponds/fields with an average area of $2.4 \times 10^4$ m² within the same station (see Fig. S1 in the supplemental material). All the irrigation and aquaculture water was pumped from the same river.

The RC ponds/fields consist of a center paddy and a surrounding ditch. The ditch is 3.0 to 4.0 m wide and 1.5 to 2.0 m deep. Nylon nets with heights of 0.3 to 0.5 m were placed around the outer banks to prevent the escape of the crayfish. In the RC model, crayfish were mainly cultured in two seasons. In the first season (March to June), about 300 kg/ha crayfish larvae (approximately 5 g of each) was added in March, and commercial feeds were fed once a day. Mature crayfish were harvested over April to June. Mid-rice (glutinous) was planted on 20 June 2019 and harvested on 25 October 2019. Before rice planting, 300 kg/ha chemical fertilizer (N:P:K, 17:13:15) was applied to the RC model without further

application of pesticides and other fertilizers for rice growth. The water depth decreased in the center field due to drainage of the paddy. Then, crayfish migrated to the ditch and mated in the ditch. During the second season (June to October), no crayfish larvae were added at the beginning due to crayfish self-propagation inside the paddies, and commercial feeds and soybean were added twice a week. Mature crayfish were harvested at the beginning of August. Compared with the first season, fewer mature crayfish were harvested in the second season. After rice harvesting, all the rice residues were returned to the paddy fields and the center fields were flooded with about 30- to 50-cm-deep water. Thereafter, crayfish returned to the paddy along with the irrigation water. Crayfish self-propagated inside the paddies, so that soybean and corn were added as supplements once a week from November to the following February.

In the CM model, crayfish were mainly cultured in two seasons as well. In the first season (March to June), the management process, such as stocking density and feeding trial, was similar to the RC model. In the second season (June to October), about 225 kg/ha crayfish larvae (approximately 5 g of each) was added in June, and mature crayfish were harvested at the beginning of August. The stocking density of crayfish was controlled by removing or adding crayfish in both the RC and CM models for all the seasons.

**Water microbial collection and physicochemical parameter determination.** Water samples were collected from each pond/field across multiple time points. At each time point, 2 L of water was taken from the surface (10 to 20 cm), middle (50 to 60 cm), and bottom (120 to 130 cm) at each sampling spot (only surface water was taken from the spot at the center paddy in the RC model), mixed into one sample (10 L), and then prefiltered with 100-$\mu$m nylon mesh to reduce impurities. Following this, 500 mL of mixed water was sequentially filtered through a 0.22-$\mu$m polycarbonate membrane (Millipore, Merck) to collect the water microbes. Two technical replicates were designed for each pond/field. The filters that contained water microbes were immediately placed in sterile 10-mL centrifuge tubes with labels for DNA extraction. All the samples were immediately stored at −80°C until DNA extraction. In total, we obtained 36 water samples from 3 CM ponds and 36 water samples from 3 RC ponds.

In addition, we determined the physicochemical data for water samples according to the method of Hou et al. (51), including water temperature (WT), pH, total nitrogen (TN), total phosphorus (TP), ammonia nitrogen ($NH_4^+$-N), and nitrite nitrogen ($NO_2$-N) (see Table S6 in the supplemental material). Briefly, WT and pH were measured using a portable logging multiparameter water quality meter (YSI Professional Plus). TN was measured using the alkaline potassium persulfate digestion-UV spectrophotometric method. TP was determined using the ammonium molybdate-UV spectrophotometry method. $NH_4^+$-N was determined using the Nessler reagent UV spectrophotometry method. $NO_2$-N was determined using the $N$-(1-naphthyl)-ethylenediamine spectrophotometry method.

**Crayfish gut microbial collection and phenotype recordation.** Crayfish were randomly recaptured, and eight healthy mature crayfish were selected and pooled into two technical replicates for each pond/field across multiple time points. To assess crayfish growth, we determined and recorded the phenotype of each crayfish, including the total length (TL), total weight (TW), abdominal length (AbL), abdominal weight (AbW), and carapace length (CL) (Table S11).

After recording the phenotype of each crayfish, we collected the gut microbes according to the method in reference 52. All crayfish dissections were performed under sterile conditions. Surgical tools were sterilized using 75% ethanol and flamed prior to use and between incisions. The body surface of each crayfish was sterilized with 75% ethanol 3 times. Then, the guts of these crayfish were collected and pooled into sterile 10-mL centrifuge tubes with labels. Next, these gut samples were ground in a sterile grinder to collect gut microbes. All the samples were immediately stored at −80°C until DNA extraction. In total, we obtained 36 crayfish gut samples from 3 RC ponds and 30 crayfish gut samples from 3 CM ponds (no gut samples were collected at time point f due to the small size of crayfish at this time point in the CM ponds).

**DNA extraction and amplicon sequencing.** The total genomic DNA for each sample was extracted using the Universal Genomic DNA kit (Omega, USA) according to the manufacturer's instructions with some modifications. The quality and quantity of isolated DNA were measured using a NanoDrop 2000 spectrophotometer (Thermo, USA). In total, 53 gut samples and 57 water samples were successful for 16S amplicon sequencing (Table S1). The V3-V4 hypervariable regions of the 16S rRNA genes were PCR amplified from a DNA aliquot of the water and gut samples using the forward primer 338F (5′-ACTCCTACGGGAGGCAGCAG-3′) and the reverse primer 806R (5′-GGACTACNNGGGTATCTAAT-3′) (53). To minimize reaction-level PCR bias, 10 ng of the purified DNA template from each sample was amplified with a 20-$\mu$L reaction system. See the supplemental material (Table S15) for details of PCR cycling conditions. 16S rRNA gene amplicon library preparation and sequencing were performed according to the manufacturer's protocol at Majorbio Bio-Pharm Technology Co., Ltd., Shanghai, China. After quality control, quantification, and normalization of the DNA libraries, 250-bp paired-end (PE) reads were generated from the Illumina MiSeq PE300 platform according to the manufacturer's instructions with modifications. More than 20,000 clean reads were generated for each amplicon sample.

**16S rRNA gene amplicon data analysis.** The raw PE reads were quality controlled by removing adaptor and primer sequences, trimming, and removing low-quality reads (reads with N bases and a minimum quality threshold of 20). The high-quality PE reads were merged using FLASH software with the default setting (54). The merged sequences were clustered into OTUs with at least 97% sequence similarity using the UPARSE pipeline (55). Chimaeras sequences were removed by applying USEARCH (56) against the SILVA database V132.0 database (57). To obtain the taxonomic information of the OTUs, representative sequences of each OTU were generated and aligned against the SILVA V132.0 database (57) using the RDP classifier (58). The OTUs and merged sequences which were defined as unknown,

chloroplast, mitochondria, plants, or animals were removed. The relative abundance tables for taxa were generated based on the read count for each taxon across samples by using the total-sum scaling (TSS) method (59). To normalize the sequencing depth, the OTU table was rarefied with the least sequences (the sequences with total number of >20, and >5 of 3 samples were held) in QIIME V1.9.1 (60) for further analyses (61). After normalization, 15,756 reads were left per sample.

**Microbial functional prediction.** Microbial function prediction was performed using the PICRUSt2 (Phylogenetic Investigation of Communities by Reconstruction of Unobserved States, V2.1.0-b) software (62). The obtained OTU abundance table was normalized by 16S rRNA copy numbers, and metagenomic function prediction was executed by predict metagenomes.py using the KEGG (Kyoto Encyclopedia of Genes and Genomes) database (63). The abundance table of KEGG orthologs (KOs) was generated based on multiple normalized OTU abundance and their predicted functional traits. The abundance table of KEGG level 2 and level 3 pathways was obtained based on the abundance of each KO and its pathway annotation.

**Comparison analysis across two breeding models.** Alpha diversity was calculated for each sample using the Shannon and Sobs index (59). The significant differences in alpha diversity between two breeding models were determined using paired Wilcoxon rank sum test (64). The taxonomic dissimilarity analysis between samples (beta diversity) was performed using the nonmetric multidimensional scaling (NMDS) with analysis of similarities (ANOSIM) method (60) based on unweighted UniFrac distances (61). The associations of crayfish phenotypes or water physicochemical features with their overall microbiomes were determined using permutational multivariate analysis of variance (PERMANOVA) with 999 permutations in the R package VEGAN (65). Based on the abundance profiles, the microbial features (OTUs, phyla, and KEGG leve1 2 and level 3 pathways and KOs) with significantly differential abundances between two breeding models were determined using paired DESeq2 (66) with a negative binomial generalized linear model. The read count matrix for DESeq2 testing was normalized using the DESeqVS method (59). The significant differences in water physicochemical features and crayfish phenotypes between two breeding models were determined using paired ANOVA and Tukey honestly significant difference (HSD) method (67). $P$ values for multiple testing were corrected using the Benjamini-Hochberg (BH) method (68). The RC-enriched (abundance significantly higher than that in the CM model) or the RC-depleted (abundance significantly lower than that in the CM model) microbes were determined based on the corrected $P < 0.05$. The relative abundances of the RC-enriched or the RC-depleted taxa and functional traits are shown using the Pheatmap package in R software (69). To demonstrate a clear RC-enriched/depleted pattern, the relative abundance of each taxon or functional trait was normalized by removing the mean (centering) and dividing by the standard deviation (scaling). Based on the $\log_2$ fold change of abundance for the enriched and depleted OTUs between the RC and CM samples, the pairwise Spearman correlation coefficient ($\rho$) values were calculated. Correlations between two items were considered statistically robust only if $|\rho|$ was $\geq 0.6$ and $P$ was $<0.05$. The cooccurrence network was visualized using Cytoscape 3.4.0 (70).

The indicator and keystone OTUs of gut and water microbiomes for the RC or CM model were determined according to the methods described by Hartman et al. (71). Briefly, the indicator OTUs were identified using both the indicspecies and edgeR methods from the R package (71). Then, we constructed cooccurrence networks for indicator species based on Spearman rank correlations between OTUs and visualized the positive, significant correlations ($\rho > 0.7$ and $P < 0.001$). The networks were visualized with the Fruchterman-Reingold layout with 100 permutations using the igraph package in R software (69). We identified keystone OTUs separately for the water and gut microbiome networks and defined them as those nodes within the top 1% of node degree values of each network. The correlations of crayfish phenotypes or water physicochemical features with microbes (the RC-enriched/depleted OTUs and indicator and keystone OTUs) were determined using Spearman's rank correlation and were shown with a color gradient denoting Spearman's correlation coefficients using a heatmap.

**Data availability.** The raw sequence data of 16S amplicon sequencing were deposited into the Sequence Read Archive of NCBI (accession number PRJNA716552).

## SUPPLEMENTAL MATERIAL

Supplemental material is available online only.

**SUPPLEMENTAL FILE 1**, PDF file, 2 MB.

**SUPPLEMENTAL FILE 2**, XLSX file, 1.8 MB.

## ACKNOWLEDGMENTS

We sincerely thank all the individuals who participated in this research.

This work was funded by the National Key Research and Development Program of China (2020YFD0900303) and the Hubei Agricultural Sciences and Technology Innovation Center (2019-620-000-001-33).

Z.G. conceived and supervised the project. L.C. and Z.G. designed the experiment. L.C., W.W., and Z.X. collected samples, extracted DNA, and performed experiments. L.C. and J.X. analyzed the data. J.X., L.C., and Z.G. wrote the manuscript. J.X., L.C., Z.G., R.H.,

Y.Z., and J.Z. revised the manuscript. All authors read and approved the final manuscript.

We declare no competing interests.

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
