## [Reviewer comments · Microbiology Spectrum]

Microbiology Spectrum

The microbiome structure of a rice–crayfish integrated breeding model and its association with crayfish growth and water quality

Ling Chen, Jin Xu, Weitao Wan, Zhiwei Xu, Ruixue Hu, Yunzeng Zhang, Jinshui Zheng, and Zemao Gu

Corresponding Author(s): Zemao Gu, Huazhong Agricultural University

Review Timeline:

Submission Date:	November 11, 2021
Editorial Decision:	December 3, 2021
Revision Received:	January 25, 2022
Editorial Decision:	February 11, 2022
Revision Received:	February 13, 2022
Accepted:	February 16, 2022

Editor: Konstantinos Kormas

Reviewer(s): Disclosure of reviewer identity is with reference to reviewer comments included in decision letter(s). The following individuals involved in review of your submission have agreed to reveal their identity: Rossanna Rodríguez-Canul (Reviewer #1); Changyou Song (Reviewer #2)

Transaction Report:

DOI: <https://doi.org/10.1128/spectrum.02204-21>

December 3, 2021

Dr. Zema Gu
Huazhong Agricultural University
Shizi mountain street
Wuhan
China

Re: Spectrum02204-21 (The microbiome structure of a rice-crayfish integrated breeding model and its association with crayfish growth and water quality)

Dear Dr. Zema Gu:

Along with your point-by-point rebuttal and revised manuscript, please include in your discussion a brief comparison of your findings compared to those from the following paper: Wang Y, Wang C, Chen Y, Zhang D, Zhao M, Li H and Guo P (2021) Microbiome Analysis Reveals Microecological Balance in the Emerging Rice-Crayfish Integrated Breeding Mode. *Front. Microbiol.* 12:669570. doi: 10.3389/fmicb.2021.669570 "

Link Not Available

Sincerely,

Konstantinos Kormas

Journals Department
Reviewer comments:

Reviewer #1 (Comments for the Author):

Review of the MS. Spectrum 02204-21. The microbiome structure of a rice-crayfish integrated breeding model and its association with crayfish growth and water quality.

This paper is describing the comparative evaluation of two types of crayfish breeding: rice-crayfish (RC) and monoculture crayfish (MC). For this the authors used gut microbiome and water microbiome. The authors also used parameters such as ammonia, nitrite, and phosphorus contents to evaluate if the system is suitable for aquaculture conditions as well as growth.

Abstract

Line 24 Change "however" for "Up to date"

Lines 26-27. Revise "and the association of the microbiome with water quality and crayfish 27 growth". It has been written in lines 23-24.

Line 32. Place the statistically significance

Line 34. Refer the organism as sp. after the genus or instead of the taxa, refer it as "the genus"

Introduction

Lines 85-95, Please, check the whole sentence, it is repeated twice.

Lines 95-97. Delete the paragraph.

Results

Lines 100-103, please re-write the paragraph: How many water and gut samples from the RC and how many water and gut samples from the MC from three landfields?. Please add a description of the letters a-f described in figure 1.

Line 108, what was the criteria used to first collect samples individually and then pooled to obtain the microbiome. Please, note that there are several differences during seasons and locations.

Line 111. Please add the developmental stages analyzed in this study. Add an additional column in the supplementary table S2 and Table 2B with the number of organisms by developmental stage in RC and MC.

Line 115. Please check this analysis, t student is a parametric test. Is alpha diversity a parametric variable?. The authors need to be aware that are comparing gut microbiota across several developmental stages (from the same period of sampling?, from different periods of sampling?. This comment is also for the water microbiota.

Lines 146-147, place either "genus" before Bacillus, etc., or sp. after each one.

Line 153. Add "sp" after the description of each genus.

Line 190-192 Water quality is part of the M & M.

Line 196-197. This statement is important.

Line 197-200 Crayfish growth is part of the M & M.

Line 252. Fiber content was not measured in this study.

Line 256. In the M & M section it was mentioned that the gut microbiome was not done.

Discussion

Lines 240-242. Delete "we compared the crayfish gut and water microbiomes between two different breeding models in order to better understand breeding model-specific microbes and their relationship with water quality and crayfish growth".

Line 243. Phyla Proteobacteria, Actinobacteria, Bacteroidetes, and Firmicutes are common in crustaceans.

Line 247. The diversity and community structure of both gut and water microbiome was different between RC and MC". Date of harvesting would have contributed to this. In none of the analyses the authors included time. Nutrients were important but what other abiotic parameters would have contributed?, sediments were also important?.

Line 254-256. This paragraph is repeated. Delete it. "Consistent with this finding, the bacterial diversity in both water and gut microbiomes was found to be higher in the RC model than in the CM model in the present study".

Line 260-263. This paragraph is quite speculative. The authors did not evaluate the presence of fiber or lignin.

Line 295-299. Basing the quality of the water only in the concentration of ammonia, nitrite and phosphorus contents is quite limited. Please discuss fully what other variables are missing?

Material and methods

Line 344. Please describe the stocking density in both RC and CM models?. Did the organisms were fed with artificial food?.

Comment: Figure 1a-f is describing the characteristics of each landscape either for water or gut microbiota collection. The authors need to describe fully the experimental design.

Please also describe the criteria used for sampling collection?, is it from the surface?, the middle? or bottom water column?.

For the water collection, the authors collected 2 l from 5 sampling points and pooled them into 10 L. then, they collected 500 ml for the biota collection. What happened with the remained 9,500 ml?. What was the criteria to follow this procedure?.

Line 347. Please describe the larval stages evaluated?. Did they evaluated crayfish growth during rice harvesting or after?.

Line 348. How did the authors assessed the growth of crayfish?, did they used recaptured methods?.

Line 349, Please describe fully the dissection process and add a reference for this methodology.

Line 356: Please, re-describe the whole paragraph for RC and MC. The authors collected 8 organisms from 5 sampling sites. That means 40 organisms x 3 cultured areas?.

Line 357. Please describe fully the stage of each crayfish. By reading the paragraph I can assume that the authors analyzed the gut microbiome of crayfish from June to October, resembling the whole cycle of crayfish?. Please describe fully the process of sampling collections with dates and stages of crayfish collections and also if mature and immature crayfishes were harvested.

Line 377. Please describe the reason for using the V4 region in this study. What is the advantage of using the V4 region vs the 16S rDNA V3-V4 region?.

Line 378. Please add the primers sequence in the M & M section.

Line 379. Add all the details of the PCR mixture and PCR conditions. Add the model of the MiSeq platform used?.

Line 390. Please add the version of the SILVA database used.

Line 384. How did the authors solve the "chimeric sequences"?

Line 391. What was the sequencing depth (in reads) to rarefied the featured table after removing mitochondria, chloroplast ASVs per sample?.

Line 411. What was the reason for using t-student for alpha-diversity?. Data is not normally distributed. Also check the reference (45) it should be 44 and then 45.

Line 423. Please add the name of the chemical tested and their normal value (if any).

Line 424. What do the authors refer as phenotypic data?, is it measurement in length?, how can it be extrapolated to age?. Where they mature?, immature?. Male and female?. In some other studies the microbiome composition is different in male and female crustaceans (<https://doi.org/10.1016/j.jembe.2018.09.002>.) (<https://doi.org/10.3390/microorganisms8091376>)

Line 426. Add a reference for the BH method and what is the rationale of the test?.

Line 451. The heatmap was constructed with the R package?.

Please add in the discussion section the reference

Wang Y, Wang C, Chen Y, Zhang D, Zhao M, Li H and Guo P (2021) Microbiome Analysis Reveals Microecological Balance in the Emerging Rice-Crayfish Integrated Breeding Mode. *Front. Microbiol.* 12:669570. doi: 10.3389/fmicb.2021.669570

Reviewer #2 (Comments for the Author):

In the present study, the authors assessed the taxonomic shifts in gut and water microbiomes and their associations with water quality and crayfish growth in RC and crayfish monoculture (CM) breeding models. The result was sound in some content, that could supply some valuable information to the literature and aquaculture practice. However, there are numerous concerns to address, especially the experimental design and grammatical assignments.

Major points

1. in the experiment design, the author hypothesize that breeding model-specific microbes have significant associations with water quality and crayfish growth. However, the dietary pattern and sediment microbes are totally different between these two models. Therefore, it is not entirely reasonable to conclude the microbes in the crayfish gut was only associated with that in the water. To clearly illustrate the crayfish growth associated microbes between these two breeding models, the authors should at least include the microbe alteration from the sediment during the experiment.

2. In Fig 7, the samples collected are not representative. In detail, the crayfish determined in stage a is larger in size than the other growth stages, this is not illogical or the actual growth of animals. Meanwhile, the crayfish in c stage under RC model is smaller than that in b stage; TL and AbL between b and c growth stage under CM condition showed opposite trends with other parameters.

3. Line 234-238, to be continued with the last comments, if the growth performance is not illogical under different stages, the correlation between crayfish growth and indicator OTUs in the water and gut microbiome is not illogical.

4. in discussion section, the authors expounded an ambiguous description on how the RC model shaped the gut microbes. In addition, the relationship between water and gut microbes, as well as their action mechanisms on crayfish growth should be well interpreted.

Minor points

This manuscript should be carefully review by a native English speaker to make it easier read.

1. Line 158, "Fig 6" should be "Fig 5"

Staff Comments:

Preparing Revision Guidelines

For complete guidelines on revision requirements, please see the journal Submission and Review Process requirements at <https://journals.asm.org/journal/Spectrum/submission-review-process>. **Submissions of a paper that does not conform to**

Microbiology Spectrum guidelines will delay acceptance of your manuscript. "

Please return the manuscript within 60 days; if you cannot complete the modification within this time period, please contact me. If you do not wish to modify the manuscript and prefer to submit it to another journal, please notify me of your decision immediately so that the manuscript may be formally withdrawn from consideration by Microbiology Spectrum.

Review of the MS. Spectrum 02204-21. The microbiome structure of a rice–crayfish integrated breeding model and its association with crayfish growth and water quality.

This paper is describing the comparative evaluation of two types of crayfish breeding: rice-crayfish (RC) and monoculture crayfish (MC). For this the authors used gut microbiome and water microbiome. The authors also used parameters such as ammonia, nitrite, and phosphorus contents to evaluate if the system is suitable for aquaculture conditions as well as growth.

Abstract

Line 24 Change “however” for “Up to date”

Lines 26-27. Revise “and the association of the microbiome with water quality and crayfish growth”. It has been written in lines 23-24.

Line 32. Place the statistically significance

Line 34. Refer the organism as sp. after the genus or instead of the taxa, refer it as “the genus”

Introduction

Lines 85-95, Please, check the whole sentence, it is repeated twice.

Lines 95-97. Delete the paragraph.

Results

Lines 100-103, please re-write the paragraph: How many water and gut samples from the RC and how many water and gut samples from the MC from three landfields?. Please add a description of the letters a-f described in figure 1.

Line 108, what was the criteria used to first collect samples individually and then pooled to obtain the microbiome. Please, note that there are several differences during seasons and locations.

Line 111. Please add the developmental stages analyzed in this study. Add an additional column in the supplementary table S2 and Table 2B with the number of organisms by developmental stage in RC and MC.

Line 115. Please check this analysis, t student is a parametric test. Is alpha diversity a parametric variable?. The authors need to be aware that are comparing gut microbiota across several developmental stages (from the same period of sampling?, from different periods of sampling?. This comment is also for the water microbiota.

Lines 146-147, place either “genus” before *Bacillus*, etc., or sp. after each one.

Line 153. Add “sp” after the description of each genus.

Line 190-192 Water quality is part of the M & M.

Line 196-197. This statement is important.

Line 197-200 Crayfish growth is part of the M & M.

Line 252. Fiber content was not measured in this study.

Line 256. In the M & M section it was mentioned that the gut microbiome was not done.

Discussion

Lines 240-242. Delete “we compared the crayfish gut and water microbiomes between two different breeding models in order to better understand breeding model-specific microbes and their relationship with water quality and crayfish growth”.

Line 243. Phyla Proteobacteria, Actinobacteria, Bacteroidetes, and Firmicutes are common in crustaceans.

Line 247. The diversity and community structure of both gut and water microbiome was different between RC and MC”. Date of harvesting would have contributed to this. In none of the analyses the authors included time. Nutrients were important but what other abiotic parameters would have contributed?, sediments were also important?.

Line 254-256. This paragraph is repeated. Delete it. “Consistent with this finding, the bacterial diversity in both water and gut microbiomes was found to be higher in the RC model than in the CM model in the present study”.

Line 260-263. This paragraph is quite speculative. The authors did not evaluate the presence of fiber or lignin.

Line 295-299. Basing the quality of the water only in the concentration of ammonia, nitrite and phosphorus contents is quite limited. Please discuss fully what other variables are missing?

Material and methods

Line 344. Please describe the stocking density in both RC and CM models?. Did the organisms were fed with artificial food?. Comment: Figure 1a-f is describing the characteristics of each landscape either for water or gut microbiota collection. The authors need to describe fully the experimental design.

Please also describe the criteria used for sampling collection?, is it from the surface?, the middle? or bottom water column?.

For the water collection, the authors collected 2 l from 5 sampling points and pooled them into 10 L. then, they collected 500 ml for the biota collection. What happened with the remained 9,500 ml?. What was the criteria to follow this procedure?.

Line 347. Please describe the larval stages evaluated?. Did they evaluated crayfish growth during rice harvesting or after?.

Line 348. How did the authors assessed the growth of crayfish?, did they used recaptured methods?.

Line 349, Please describe fully the dissection process and add a reference for this methodology.

Line 356: Please, re-describe the whole paragraph for RC and MC. The authors collected 8 organisms from 5 sampling sites. That means 40 organisms x 3 cultured areas?.

Line 357. Please describe fully the stage of each crayfish. By reading the paragraph I can assume that the authors analyzed the gut microbiome of crayfish from June to October, resembling the whole cycle of crayfish?. Please describe fully the process of sampling collections with dates and stages of crayfish collections and also if mature and immature crayfishes were harvested.

Line 377. Please describe the reason for using the V4 region in this study. What is the advantage of using the V4 region vs the 16S rDNA V3-V4 region?.

Line 378. Please add the primers sequence in the M & M section.

Line 379. Add all the details of the PCR mixture and PCR conditions. Add the model of the MiSeq platform used?.

Line 390. Please add the version of the SILVA database used.

Line 384. How did the authors solve the “chimeric sequences”?

Line 391. What was the sequencing depth (in reads) to rarefied the featured table after removing mitochondria, chloroplast ASVs per sample?.

Line 411. What was the reason for using t-student for alpha-diversity?. Data is not normally distributed. Also check the reference (45) it should be 44 and then 45.

Line 423. Please add the name of the chemical tested and their normal value (if any).

Line 424. What do the authors refer as phenotypic data?, is it measurement in length?, how can it be extrapolated to age?. Where they mature?, immature?. Male and female?. In some other studies the microbiome composition is different in male and female crustaceans (<https://doi.org/10.1016/j.jembe.2018.09.002>.) (<https://doi.org/10.3390/microorganisms8091376>)

Line 426. Add a reference for the BH method and what is the rationale of the test?.

Line 451. The heatmap was constructed with the R package?.

Please add in the discussion section the reference

Wang Y, Wang C, Chen Y, Zhang D, Zhao M, Li H and Guo P (2021) Microbiome Analysis Reveals Microecological Balance in the Emerging Rice–Crayfish Integrated Breeding Mode. *Front. Microbiol.* 12:669570. doi: 10.3389/fmicb.2021.669570

Responds to the reviewer's comments:

We really appreciate your professional and constructive suggestions. We have studied your comments carefully and have made revisions which were highlight by yellow color in the revised manuscript. Besides, we consulted professional English editor to improve our writing, and the editorial certificate was attached to the end of this letter. The followings are our responses to your comments point by point.

Editor:

Please include in your discussion a brief comparison of your findings compared to those from the following paper: Wang Y, Wang C, Chen Y, Zhang D, Zhao M, Li H and Guo P (2021) Microbiome Analysis Reveals Microecological Balance in the Emerging Rice-Crayfish Integrated Breeding Mode. *Front. Microbiol.* 12:669570. doi: 10.3389/fmicb.2021.669570

Response: Thank you for your good suggestions. We have added a comparison of our results compared to the paper “Microbiome Analysis Reveals Microecological Balance in the Emerging Rice-Crayfish Integrated Breeding Mode” in the discussion section as follows. “In the present study, we found that several bacterial phyla, such as Proteobacteria, Actinobacteria, Bacteroidetes, and Firmicutes, were dominant in both gut and water microbiomes of two breeding models across multi-time points; these findings are similar to previous findings in microbiomes of rearing water and guts of other crustaceans (16-20). Compared to the CM model, the diversity and community structure of both gut and water microbiomes were significantly differed in the RC model, suggesting that the breeding model may affect the microbiome assembly of crayfish gut and water. However, these findings were not consistent with the previous study, in which water, sediment, and crayfish samples from single time point were collected, and no statistically significant differences of microbiome diversity and

structure between the RC and CM models were found (19). Due to the different environment status of the RC model at different times, we supposed that the sampling time could be a reasonable factor cause of the different results. Amounts of studies have reported a variable microbiome structure at different sampling times (21-23). Also, our results showed a fluctuating diversity and structure of both water and gut microbiomes across six sampling times. Therefore, we used multi-time points sampling strategy to comprehensively capture the microbiome for the RC and CM models (Page 10, lines 234-250).

Reviewer #1:

This paper is describing the comparative evaluation of two types of crayfish breeding: rice-crayfish (RC) and monoculture crayfish (MC). For this the authors used gut microbiome and water microbiome. The authors also used parameters such as ammonia, nitrite, and phosphorus contents to evaluate if the system is suitable for aquaculture conditions as well as growth.

Response: Thank you for excellent comments.

Abstract

Line 24 Change “however” for “Up to date”

Response: Thank you for your good suggestion, we have revised it on line 23.

Lines 26-27. Revise “and the association of the microbiome with water quality and crayfish growth”. It has been written in lines 23-24.

Response: Thank you for your good suggestion, we have revised it.

Line 32. Place the statistically significance

Response: Thank you for your good suggestion, we have added the p-value on line 31.

Line 34. Refer the organism as sp. after the genus or instead of the taxa, refer it as “the genus”

Response: Thank you for your good suggestion, we have added the “sp.” for each genus.

Introduction

Lines 85-95, Please, check the whole sentence, it is repeated twice.

Response: Thank you for your good suggestion, we rewrote this paragraph to make our hypothesis, approaches, and aims clearer (Page 4, lines 85-96).

Lines 95-97. Delete the paragraph.

Response: Thank you for your good suggestion, we rewrote this paragraph to make our hypothesis, approaches, and aims clearer (Page 4, lines 85-96).

3, Response to comment: Results

Lines 100-103, please re-write the paragraph: How many water and gut samples from the RC and how many water and gut samples from the MC from three land fields? Please add a description of the letters a-f described in figure 1.

Response: Thank you for your good suggestion, we have added the detailed samples information in this part (Page 30, lines 100-101). The description of the letters a-f has been added in the introduction and figure legend (Page 30, lines 772-777).

Line 108, what was the criteria used to first collect samples individually and then pooled to obtain the microbiome. Please, note that there are several differences during seasons and locations.

Response: There are many studies have been suggested that microbiome was dynamic across time. Consist with this, there are several differences during seasons you mentioned. To make samples in our study were much more representative and comprehensive of two breeding models, we collected samples across multi-time

points, which should be better than the sample of single time point. Furthermore, to make our study was much focused and easily understand, we mainly focused on the comparison between breeding models, but not among different time points. Therefore, we focused on the description of the microbiome structure between two breeding models (pooled all the samples from multi-time point for each model, and each time point between two models).

Line 111. Please add the developmental stages analyzed in this study. Add an additional column in the supplementary table S2 and Table 2B with the number of organisms by developmental stage in RC and MC.

Response: Thank you for your great suggestion. However, to make **our study much focused and easy understanding**, we mainly focused on the comparison between breeding models, but not among different time points. Therefore, we focused on the description of the microbiome structure between two breeding models. In addition, the time points were related to **rice growth but not the crayfish developmental stages**. For all the time points, we only measured the mature crayfish. In the next step, we will focus on the microbiome study across developmental stages of crayfish in two breeding models.

Line 115. Please check this analysis, t student is a parametric test. Is alpha diversity a parametric variable? The authors need to be aware that are comparing gut microbiota across several developmental stages (from the same period of sampling? from different periods of sampling? This comment is also for the water microbiota.

Response: Thanks for your critical suggestions. We have replaced the methods of t-student to paired Wilcoxon rank-sum test as to satisfy the data of both gut microbiome and water microbiome. To rule out the differences contributed by time points, we used the paired statistical test (comparison between two breeding models

from the same period of sampling) to do the comparison between two breeding models for both gut and water microbiomes.

Lines 146-147, place either “genus” before *Bacillus*, etc., or sp. after each one.

Response: Thank you for your good suggestion, we have added the “sp.” for each genus.

Line 153. Add “sp” after the description of each genus.

Response: Thank you for your good suggestion, we have added the “sp.” for each genus.

Line 190-192 Water quality is part of the M & M.

Response: Thank you for your good suggestion, we have removed this content.

Line 196-197. This statement is important.

Response: Thank you for your comments. We have discussed the possible reasons for the difference of nitrogen between two breeding models in the discussion part. In the RC model, more nitrogen sources are required for rice growth. Besides artificial nitrogen sources, microbes can help plants to generate nitrogen sources from the environment. Accordingly, some members of Cyanobacteria involved in nitrogen fixation were found to be enriched in the water microbiome in the RC model. In addition, some members of Chloroflexi involved in nitrification and *Bacillus* sp. involved in ammonification had higher abundances in the water microbiome in the RC model than in the CM model.

Line 197-200 Crayfish growth is part of the M & M.

Response: Thank you for your good suggestion, we have removed this content.

Discussion

Line 252. Fiber content was not measured in this study.

Response: Thank you for your comments. We do not measure fiber content in this study due to funding limitation. However, several previous studies have reported higher fiber content in the RC model than in the CM model. Here, we discussed the possible link between fiber content and microbes in the RC model. Some members of Actinobacteria that contribute significantly to the turnover of complex biopolymers and decomposition of organic matter had higher abundances in the RC model than in the CM model. Moreover, gut microbes positively correlated with fiber intake, such as *Ruminococcaceae* sp., *Bifidobacteria* sp., and lactic acid bacteria, were more abundant in the RC model than in the CM model. To confirm those links between fiber content and microbes in the RC model, we need conduct much more experiments, such as fiber content measure, culture of target microbes, and synthetic communities in the next step.

Line 256. In the M & M section it was mentioned that the gut microbiome was not done.

Response: Thank you for your comments. The gut microbiome at time point f in RC model was done, which has been described in M & M section as following. In total, we obtained 36 crayfish gut samples from 3 RC ponds and 30 crayfish gut samples from 3 CM ponds (no gut samples were collected at time point f due to the small size of crayfish at this time point in CM pond) (Page 17, lines 419-421).

Lines 240-242. Delete “we compared the crayfish gut and water microbiomes between two different breeding models in order to better understand breeding model-specific microbes and their relationship with water quality and crayfish growth”.

Response: Thank you for your good suggestion. We have removed this content.

Line 243. Phyla Proteobacteria, Actinobacteria, Bacteroidetes, and Firmicutes are common in crustaceans.

Response: Thank you for your comments. Absolutely, we totally agreed with that, as we have mentioned “these findings are similar to previous findings in microbiomes of rearing water and guts of other crustaceans” on page 10, lines 237-238.

Line 247. “The diversity and community structure of both gut and water microbiome was different between RC and MC”. Date of harvesting would have contributed to this. In none of the analyses the authors included time. Nutrients were important but what other abiotic parameters would have contributed? sediments were also important?

Response: Thanks for your critical suggestions. To rule out the differences contributed by time points, we used the paired statistical test (comparison between two breeding models from the same period of sampling) to do the comparison between two breeding models for both gut and water microbiomes. To make our study **much focused and easily understand**, we mainly focused on the comparison between breeding models, but not among different time points. We agreed with you that many abiotic parameters, such as sediments would contribute the microbiome differences between two breeding models. Due to funding limitation and to make our study **much focused and easy understanding**, we do not collect the microbial and physicochemical data of sediments in this study. To confirm causative roles between abiotic factors, such as nutrient and sediment, and microbiome in the RC model, the further studies, including abiotic factor measure, culture of target microbes, and synthetic communities need to be conducted in the future.

Line 254-256. This paragraph is repeated. Delete it. “Consistent with this finding, the bacterial diversity in both water and gut microbiomes was found to be higher in the RC model than in the CM model in the present study”.

Response: Thank you for your good suggestions. We have revised it.

Line 260-263. This paragraph is quite speculative. The authors did not evaluate the presence of fiber or lignin.

Response: Thank you for your comments. We do not measure fiber content in this study due to funding limitation. However, several previous studies have reported higher fiber content in the RC model than in the CM model. Here, we discussed the possible link between fiber content and microbes in the RC model. Some members of Actinobacteria that contribute significantly to the turnover of complex biopolymers and decomposition of organic matter had higher abundances in the RC model than in the CM model. Moreover, gut microbes positively correlated with fiber intake, such as *Ruminococcaceae* sp., *Bifidobacteria* sp., and lactic acid bacteria, were more abundant in the RC model than in the CM model. To confirm those links between fiber content and microbes in the RC model, we need to conduct much more experiments, such as fiber content measure, culture of target microbes, and synthetic communities in the next step.

Line 295-299. Basing the quality of the water only in the concentration of ammonia, nitrite and phosphorus contents is quite limited. Please discuss fully what other variables are missing?

Response: Thanks for your good suggestion. In this study, we measured some of the most important water quality parameters, including water temperature, pH, ammonia, nitrite, nitrogen, and phosphorus due to funding limitation. We found that the concentrations of ammonia, nitrite and phosphorus were different between two

breeding model. Therefore, we did the discussion regarding the possible links between microbiome and these water quality parameters. Following your suggestion, we discussed the RC-enriched microbes, which can modulate other water quality parameters, including transparency, total dissolved solids, pH, conductivity, chemical oxygen demand, dissolved oxygen, biological oxygen demand, alkalinity, and hardness. In the future, we need to collect water quality parameters that missed in this study and target microbes to further confirm our findings.

Material and methods

Line 344. Please describe the stocking density in both RC and CM models?

Response: Thank you for your suggestion. We added the stocking density information in the M&M part. For the first season of crayfish cultivation (March-June), about 300 kg/ha crayfish larvae were added in both the RC and CM models at the beginning. During the first season, the management process, such as stocking density and feeding trial were similar between these two breeding models. For the second season of crayfish cultivation (June-October), no larva was added in the RC model due to crayfish self-propagated inside the paddies, while about 225 kg/ha larvae were added in the CM model at the beginning. The stocking density of crayfish was controlled by removing or adding crayfishes both in RC and CM models for all the season.

Did the organisms were fed with artificial food?

Response: Yes, in the first season, commercial feeds were feeding once a day for both models, while commercial feeds and soybean were added twice a week in the second season for both models.

Comment: Figure 1a-f is describing the characteristics of each landscape either for water or gut microbiota collection. The authors need to describe fully the experimental design.

Response: Thank you for your suggestion. we have described the complete experimental design in the M & M section (Pages 15-16, lines 351-384).

Please also describe the criteria used for sampling collection? is it from the surface? the middle? or bottom water column? For the water collection, the authors collected from 5 sampling points and pooled them into 10 L. then, they collected 500 ml for the biota collection. What happened with the remained 9,500 ml? What was the criteria to follow this procedure?

Response: Thank you for your comments. To increase the representativeness of each pond, a total of 10 L water was collected from 5 sampling spots (surface, middle, and bottom for each sample site). Then we pre-filtered with 100- μ m nylon mesh to reduce impurities. After that, 500 ml of them were used for the biota collection, 500 mL of them were used for determining water physicochemical parameters, and the remained 9,000 ml were discarded.

Line 347. Please describe the larval stages evaluated? Did they evaluated crayfish growth during rice harvesting or after?

Response: Thank you for your suggestions. In this study, we only collected the mature crayfishes and measured their growth-related data at each time point. The time points were related to rice growth but the crayfish developmental stages. To make our study much focused and easily understand, we mainly focused on the comparison between breeding models, but not among different time points. In the next step, we will focus on the microbiome study across developmental stages of crayfish in two breeding models.

Line 348. How did the authors assessed the growth of crayfish? did they used recaptured methods?

Response: Yes, we do used recaptured methods. Crayfishes were randomly recaptured, and eight healthy mature crayfishes were selected and pooled into two technical replicates for each pond/field across multi-time points. To assess crayfish growth, we determined and recorded the phenotype of each crayfish, including the total length (TL), total weight (TW), abdominal length (AbL), abdominal weight (AbW), and carapace length (CL) of each crayfish (Table S11) (Page 17, lines 407-411).

Line 349, Please describe fully the dissection process and add a reference for this methodology.

Response: Thanks for your good suggestion. We re-described the whole dissection process and added a reference for this methodology. The dissection process was described as follows: “After recorded the phenotype of each crayfish, we collected the gut microbes according to the reference (52). All crayfish dissections were performed under sterile conditions. Surgical tools were sterilized using 75% ethanol and flamed prior to use and between incisions. Sterilizing the body surface of each crayfish with 75% ethanol 3 times. Then the guts of these crayfishes were collected and pooled into sterile 10 mL centrifuge tubes with labels. Next, these guts samples were ground in a sterile grinder to collect gut microbes, and divided into two technical replicates. All the samples were immediately stored at -80°C until DNA extraction.” (Page 17, lines 412-421).

Line 356: Please, re-describe the whole paragraph for RC and MC. The authors collected 8 organisms from 5 sampling sites. That means 40 organisms x 3 cultured areas?

Response: Thanks for your good suggestion. We have re-described the whole paragraph for RC and MC in the section “Crayfish gut microbial collection and

phenotype recordation” on page 17, lines 407-409. At each time point, about 8 healthy mature crayfishes were randomly recaptured from each pond/field instead of from 5 sampling sites.

Line 357. Please describe fully the stage of each crayfish. By reading the paragraph I can assume that the authors analyzed the gut microbiome of crayfish from June to October, resembling the whole cycle of crayfish? Please describe fully the process of sampling collections with dates and stages of crayfish collections, and also if mature and immature crayfishes were harvested.

Response: Thank you for your professional suggestions, we have re-described the whole paragraph for the process of sampling collections with dates and stages of crayfish collections in the section “Crayfish gut microbial collection and phenotype recordation”. 1) As we all known the major difference between the RC and CM model is that the rice growing in the RC model. Based on this, we taking samples from June to October according to the rice developmental stages, instead of the cycle of crayfish. At the same time, the second season of crayfish cultivation is started from June to October. 2) The details and fully process of sampling collections with dates and stages of crayfish collections were re-described as follows (Page 17, lines 406-421): “Crayfishes were randomly recaptured, and eight healthy mature crayfishes were selected and pooled into two technical replicates for each pond/field across multi-time points. To assess crayfish growth, we determined and recorded the phenotype of each crayfish, including the total length (TL), total weight (TW), abdominal length (AbL), abdominal weight (AbW), and carapace length (CL) (Table S11). After recorded the phenotype of each crayfish, we collected the gut microbes according to the reference. All crayfish dissections were performed under sterile conditions. Surgical tools were sterilized using 75% ethanol and flamed prior to use and between incisions.

Sterilizing the body surface of each crayfish with 75% ethanol 3 times. Then the guts of these crayfishes were collected and pooled into sterile 10 mL centrifuge tubes with labels. Next, these guts samples were ground in a sterile grinder to collect gut microbes. All the samples were immediately stored at -80°C until DNA extraction. In total, we obtained 36 crayfish gut samples from 3 RC ponds and 30 crayfish gut samples from 3 CM ponds (no gut samples were collected at time point f due to the small size of crayfish at this time point in CM pond).”

Line 377. Please describe the reason for using the V4 region in this study. What is the advantage of using the V4 region vs the 16S rDNA V3-V4 region?

Response: Thank you for your comments. We used the 16S rDNA V3-V4 region in this study, and we have revised this in the M & M section (Page 18, line 427-428).

Line 378. Please add the primers sequence in the M & M section.

Response: Thank you for your professional suggestions. We used the forward primer 338F (5'-ACTCCTACGGGAGGCAGCAG-3') and the reverse primer 806R (5'-GGACTACNNGGGTATCTAAT-3'), which have been added on page 18, lines 429-431.

Line 379. Add all the details of the PCR mixture and PCR conditions. Add the model of the MiSeq platform used?

Response: Thank you for your professional suggestions. The PCR mixture and PCR conditions were added in supplemental table S15 and table S16 in details. Besides, the amplicon sequencing was based on the platform Illumina Miseq PE 300, which have been added on page 18, line 437.

Table S15. PCR mixture to generated the 16S amplicons for high-throughput sequencing

Reagent	Content
5×FastPfu Buffer	4 μL
2.5 mM dNTPs	2 μL

Forward Primer (5 μ M)	0.8 μ L
Reverse Primer (5 μ M)	0.8 μ L
FastPfu PoLymerase	0.4 μ L
BSA	0.2 μ L
TempLate DNA	10 ng
Add ddH ₂ O to 20 μ L	

Table S16. PCR cycling conditions used to generated the 16S amplicons for high-throughput sequencing.

Step	Temperature ($^{\circ}$ C)	Time	Cycles
1	95	3min	1 \times
2	95	30sec	30 \times
3	55	30sec	
4	72	45sec	
5	72	10min	1 \times
6	10	hold	

Line 390. Please add the version of the SILVA database used.

Response: Thank you for your good suggestions We added the version of the SILVA database used (Silva V132.0) on page 18, line 446.

Line 384. How did the authors solve the “chimeric sequences”?

Response: Chimaeras sequences were removed by applying USEARCH against the SILVA database V132.0 database (page 18, line 445-447).

Line 391.What was the sequencing depth (in reads) to rarefied the featured table after removing mitochondria, chloroplast ASVs per sample?

Response: 15657 reads/per sample were left after rarefied with removing mitochondria, chloroplast ASVs (page 19, line 455).

Line 411. What was the reason for using t-student for alpha-diversity? Data is not normally distributed. Also check the reference (45) it should be 44 and then 45.

Response: Thank you for your good suggestions. Now we have updated the methods of t-student into paired Wilcoxon rank sum test as to satisfied the data. In addition, all the references in the present manuscript have been checked and revised.

Line 423. Please add the name of the chemical tested and their normal value (if any).

Response: Thank you for your good suggestions. The name of the chemical tested were added in the “Water microbial collection and physicochemical parameters determination” section as following (page 16, lines 397-400). “We determined the physicochemical data for water samples according to the method of Hou et al. (51), including water temperature (WT), pH, total nitrogen (TN), total phosphorus (TP), ammonia nitrogen ($\text{NH}_4^+\text{-N}$), and nitrite nitrogen ($\text{NO}_2^-\text{-N}$) (Table S6)”. And the value of those was showed in Table S 10 and Fig S7.

Line 424. What do the authors refer as phenotypic data? is it measurement in length? how can it be extrapolated to age? Where they mature? immature? Male and female? In some other studies the microbiome composition is different in male and female crustaceans

(<https://doi.org/10.1016/j.jembe.2018.09.002>.)

(<https://doi.org/10.3390/microorganisms8091376>)

Response: Thank you for your good suggestions. To assess the growth of each recaptured mature crayfish, we determined and recorded the phenotype of each, including the total length (TL), total weight (TW), abdominal length (AbL), abdominal weight (AbW), and carapace length (CL) (Table S11). Crayfishes were randomly recaptured, and eight healthy mature crayfishes were selected and pooled into two technical replicates for each pond/field across multi-time points. They included both male and female crayfishes in the RC and CM models, suggested that the potential differences caused by sex could be eliminated.

Line 426. Add a reference for the BH method and what is the rationale of the test?

Response: Thank you for your good suggestions. We have quoted a reference about Benjamini–Hochberg (BH) method on page 20, line 484. The BH method minimized the false positive event estimation probability for multiple tests. The p-value obtained

by correction is also called the best FDR (false discovery rate). Smaller FDR indicates that smaller the probability and estimated probability of false positive events.

Line 451. The heatmap was constructed with the R package?

Response: The heatmap was constructed with the Pheatmap package in R software, which have been written on page 20, line 488.

Please add in the discussion section the reference

Wang Y, Wang C, Chen Y, Zhang D, Zhao M, Li H and Guo P (2021) Microbiome Analysis Reveals Microecological Balance in the Emerging Rice–Crayfish Integrated Breeding Mode. *Front. Microbiol.* 12:669570. doi: 10.3389/fmicb.2021.669570

Response: Thank you for your good suggestions. We have added a comparison of our results compared to the paper “Microbiome Analysis Reveals Microecological Balance in the Emerging Rice-Crayfish Integrated Breeding Mode” in the discussion section as follows (page 10, lines 234-250). “In the present study, we found that several bacterial phyla, such as Proteobacteria, Actinobacteria, Bacteroidetes, and Firmicutes, were dominant in both gut and water microbiomes of two breeding models across multi-time points; these findings are similar to previous findings in microbiomes of rearing water and guts of other crustaceans (16-20). Compared to the CM model, the diversity and community structure of both gut and water microbiomes were significantly differed in the RC model, suggesting that the breeding model may affect the microbiome assembly of crayfish gut and water. However, these findings were not consistent with the previous study, in which water, sediment, and crayfish samples from single time point were collected, and no statistically significant differences of microbiome diversity and structure between the RC and CM models were found (19). Due to the different environment status of the RC model at different times, we supposed that the sampling time could be a reasonable factor cause of the

different results. Amounts of studies have reported a variable microbiome structure at different sampling times (21-23). Also, our results showed a fluctuating diversity and structure of both water and gut microbiomes across six sampling times. Therefore, we used multi-time points sampling strategy to comprehensively capture the microbiome for the RC and CM models.

Reviewer #2:

1. In the experiment design, the author hypothesize that breeding model specific microbes have significant associations with water quality and crayfish growth. However, the dietary pattern and sediment microbes are totally different between these two models. Therefore, it is not entirely reasonable to conclude the microbes in the crayfish gut was only associated with that in the water. To clearly illustrate the crayfish growth associated microbes between these two breeding models, the authors should at least include the microbe alteration from the sediment during the experiment

Response: Thank you for your comments. We agreed with you that dietary pattern and sediment composition were different between these two models due to the rice planting in RC model. The dietary pattern difference between two models have been confirmed in previous study “Feeding habits of *Procambarus clarkii* and food web structure in two different aquaculture system” (in Chinese with English abstract, Doi: 10.7541/2020.016). The sediment composition could be different due to the rotting of rice residues and application of artificial fertilizer. In current study, we also found the growth data of crayfish and physicochemical data of water were different between these two models. Based on those differences, we compared the microbiome composition of crayfish gut and water between two models, and generated the breeding model specific microbes. We then found that some of those breeding model specific microbes of crayfish gut and water showed strong association with crayfish

growth and water physicochemical features, respectively. In current study, we did not explore the association between gut microbes and water microbes, which has been studied in previous study “Microbiome Analysis Reveals Microecological Balance in the Emerging Rice-Crayfish Integrated Breeding Mode” (Doi: 10.3389/fmicb.2021.669570). Due to funding limitation and to make our study much focused and easily understand, we do not collect the microbial and physicochemical data of sediments in this study. In the future, we would like to study the microbial and physicochemical data of sediments to further confirm our findings in this study.

2. In Fig 7, the samples collected are not representative. In detail, the crayfish determined in stage a is larger in size than the other growth stages, this is not illogical or the actual growth of animals. Meanwhile, the crayfish in c stage under RC model is smaller than that in b stage; TL and AbL between b and c growth stage under CM condition showed opposite trends with other parameters.

Response: Thank you for your comments. In the “Material and Method” section, we have described the crayfish cultivation and our experimental design in details (page 351-384). In current study, the sampling time points were associated with rice growth but not crayfish development. In addition, we only captured the mature crayfish for each time point for both breeding models. Therefore, the crayfish growth data was not comparable among time points within same model and not associated with crayfish development. The crayfish growth data at each time point was independent. Moreover, to make our study much focused and easily understand, we mainly focused on the comparison between breeding models, but not among different time points within same model.

3. Line 234-238, to be continued with the last comments, if the growth performance is not illogical under different stages, the correlation between crayfish growth and indicator OTUs in the water and gut microbiome is not illogical.

Response: Thank you for your comments. In the Material and Method section, we have described the crayfish cultivation and our experimental design in details (page 351-384). In current study, the sampling time points were associated with rice growth but not crayfish development. In addition, we only captured the mature crayfish for each time point for both breeding models. Therefore, the growth data of crayfish was not comparable among time points within same model and not associated with crayfish development. The growth data of crayfish at each time point was independent. To make sure the statistical power and reliability of correlation, we collected the data across multi-time points to assess the correlations of crayfish gut microbes with crayfish growth data.

4. in discussion section, the authors expounded an ambiguous description on how the RC model shaped the gut microbes. In addition, the relationship between water and gut microbes, as well as their action mechanisms on crayfish growth should be well interpreted.

Response: Thank you for your comments. In this study, we mainly focused on the comparison of gut and water microbiomes between two models, and the breeding model specific microbes and their association with crayfish growth and water quality. The relationship of the microbes between the crayfish gut and water in these two breeding models was interesting but has been studied in previous study “Microbiome Analysis Reveals Microecological Balance in the Emerging Rice-Crayfish Integrated Breeding Mode” (Doi: 10.3389/fmicb.2021.669570). We have discussed the possible factors, such as nutrient, nitrogen, and sediment for shaping the gut and water

microbiomes. In addition, the relationship between water physiochemical data and water microbiome, as well as the relationship between crayfish growth and gut microbiome were interpreted in discussion section.

5. This manuscript should be carefully review by a native English speaker to make it easier read.

Response: Thank you for your great suggestion. We consulted professional English editor to improve our manuscript, and the editorial certificate was attached to the end of this letter.

6. Line 158, “Fig 6” should be “Fig 5”

Response: Thank for your careful review. We have revised it on line 157.

EDITORIAL CERTIFICATE

Date: Dec 14, 2021

Manuscript Author(s): Ling Chen, Jin Xu, Weitao Wan, Zhiwei Xu, Ruixue Hu, Yunzeng Zhang, Jinshui Zheng, Zemao Gu

Manuscript Title: The microbiome structure of a rice–crayfish integrated breeding model and its association with crayfish growth and water quality

To Whom It May Concern:

This letter confirms that the manuscript corresponding to the information detailed above was edited by a professional, native English-speaking editor at Wordvice.

We guarantee 100% language accuracy in the text, as edited and delivered to the author(s) on the date below. We make no claims as to the substantive matter covered by the paper and have not altered the intent or research content drafted by the author(s).

The author(s) may accept or reject any of our comments or suggestions upon receipt of the document we edited. Should you have any questions or concerns, please contact Wordvice at edit@wordvice.com

Sincerely,
Wordvice

Wordvice is a premier international English editing service. Our mission is to help researchers, scholars, students, and professionals reach their full potential through clear communication in their writing. By providing premium English editing services at affordable rates to clients from around the world, Wordvice seeks to tear down language barriers and contribute to the advancement of research and education.

Signature

Kevin Heintz
Managing Editor, Wordvice

Date of Issue

Dec 14, 2021

February 11, 2022

Dr. Zema Gu
Huazhong Agricultural University
Shizi mountain street
Wuhan
China

Re: Spectrum02204-21R1 (The microbiome structure of a rice-crayfish integrated breeding model and its association with crayfish growth and water quality)

Dear Dr. Zema Gu:

Link Not Available

Sincerely,

Konstantinos Kormas

Journals Department
Reviewer comments:

Reviewer #1 (Public repository details (Required)):

Authors have submitted the information to a public repository

Reviewer #1 (Comments for the Author):

Revision of Spectrum02204-21R1

The microbiome structure of a rice-crayfish integrated breeding model and its association with crayfish growth and water quality

The quality of the paper has improved and with data presented here, the authors were able to evaluate the effect of the breeding model on microbiome assembly and composition. Ans were able to associate the variability in the microbiome with water quality and crayfish growth. However, there are still some issues addressed below that need to be taken.

Lines 30 & 108, probably in a cohort of time instead of before multi-times?.

Line 89. The following statement is very important for the developing of the study. This must be included in the abstract, and results. "We collected crayfish and water samples between two breeding models across six time points of rice growth, including seedling (a), tillering and jointing (b), blooming (c), filling (d), fruiting (e), and rotting of rice residues".

This is because, there were statistical significant differences in growth and microbiome assembly in group "a" compared to the other groups.

Lines 155-158. The paragraph needs edition to make it clear. "Furthermore, we identified the indicator OTUs of water and gut microbiomes in the RC and CM models, and keystone taxa of water and crayfish gut microbiome were identified as well"

Line 190. The paragraph needs edition "significantly differed at different degrees"?. This is a correlation analysis, the results are reported in % not in degrees.

Line 191. Please describe more "across multi-time points". This is related to the six sampling collections addressed during the different stages of the rice harvesting (a-f).

Line 195. Place (tillering and jointing) after "a"

Line 262. Delete "a"

Reviewer #2 (Comments for the Author):

The authors have addressed the comments that I concern.

Staff Comments:

Preparing Revision Guidelines

Please return the manuscript within 60 days; if you cannot complete the modification within this time period, please contact me. If you do not wish to modify the manuscript and prefer to submit it to another journal, please notify me of your decision immediately so that the manuscript may be formally withdrawn from consideration by Microbiology Spectrum.

The microbiome structure of a rice–crayfish integrated breeding model and its association with crayfish growth and water quality

The quality of the paper has improved and with data presented here, the authors were able to evaluate the effect of the breeding model on microbiome assembly and composition. Authors were able to associate the variability in the microbiome with water quality and crayfish growth. However, there are some issues addressed below that need to be taken.

Lines 30 & 108, probably in a cohort of time instead of before multi-times?.

Line 89. The following statement is very important for the developing of the study. This must be included in the abstract, and results. “We collected crayfish and water samples between two breeding models across six time points of rice growth, including seedling (a), tillering and jointing (b), blooming (c), filling (d), fruiting (e), and rotting of rice residues”.

This is because, there were statistical significant differences in growth and microbiome assembly in group “a” compared to the other groups.

Lines 155-158. The paragraph needs edition to make it clear. “Furthermore, we identified the indicator OTUs of water and gut microbiomes in the RC and CM models, and keystone taxa of water and crayfish gut microbiome were identified as well”

Line 190. The paragraph needs edition “significantly differed at different degrees”?. This is a correlation analysis, the results are reported in % not in degrees.

Line 191. Please describe more “across multi-time points”. This is related to the six sampling collections addressed during the different stages of the rice harvesting (a-f).

Line 195. Place (tillering and jointing) after “a”

Line 262. Delete “a”

Responds to the reviewer's comments:

We have studied the comments carefully and made revisions which were highlight by yellow color in the revised manuscript. These changes will not influence the content and framework of the paper. We appreciate for your warm work earnestly, and hope that the correction will meet with approval. The followings are our responses to comments point by point.

Reviewer #1:

The quality of the paper has improved and with data presented here, the authors were able to evaluate the effect of the breeding model on microbiome assembly and composition. Ans were able to associate the variability in the microbiome with water quality and crayfish growth. However, there are still some issues addressed below that need to be taken.

Response: Thank you for excellent comments.

1. Lines 30 & 108, probably in a cohort of time instead of before multi-times?

Response: Thank you for your suggestion. We have revised that on lines 32, 110, and 239.

2. Line 89. The following statement is very important for the developing of the study. This must be included in the abstract, and results. "We collected crayfish and water samples between two breeding models across six time points of rice growth, including seedling (a), tillering and jointing (b), blooming (c), filling (d), fruiting (e), and rotting of rice residues". This is because, there were statistical significant differences in growth and microbiome assembly in group "a" compared to the other groups.

Response: Thank you for excellent suggestion. The statement has been added in the abstract and results section on lines 28-30 and 192-194.

3. Lines 155-158. The paragraph needs edition to make it clear. "Furthermore, we identified the indicator OTUs of water and gut microbiomes in the RC and CM models, and keystone taxa of water and crayfish gut microbiome were identified as well"

Response: Thank you for your great suggestion. We have revised this sentence as follows (lines 156-157): "Furthermore, we identified the indicator and keystone OTUs of water and gut microbiomes in the RC and CM models (Fig. 5, Table S7, Table S8, and Table S9)."

4. Line 190. The paragraph needs edition "significantly differed at different degrees"?. This is a correlation analysis, the results are reported in % not in degrees.

Response: Thank you for your question. In this paragraph, we described the significant differences of water physicochemical data between two breeding models. Thus, we considered that describe as "significantly differed at different degrees" is reasonable.

5. Line 191. Please describe more "across multi-time points". This is related to the six sampling collections addressed during the different stages of the rice harvesting (a-f).

Response: Thank you for your great suggestion. We have redescribed "across multi-time points" into "across six time points of rice growth, including seedling (a), tillering and jointing (b), blooming (c), filling (d), fruiting (e), and rotting of rice residues" on lines 192-194.

6. Line 195. Place (tillering and jointing) after "a"

Response: Thank you for your suggestion. We have revised that on line 197.

7. Line 263. Delete "a"

Response: Thank you for your suggestion. This mistake has been revised on line 265.

We would like to express our great appreciation again to you for comments on our paper.

Reviewer #2:

The authors have addressed the comments that I concern.

Response: We would like to express our great appreciation to you for comments on our paper.

February 16, 2022

Dr. Zema Gu
Huazhong Agricultural University
Shizi mountain street
Wuhan
China

Re: Spectrum02204-21R2 (The microbiome structure of a rice-crayfish integrated breeding model and its association with crayfish growth and water quality)

Dear Dr. Zema Gu:

Your manuscript has been accepted, and I am forwarding it to the ASM Journals Department for publication. You will be notified when your proofs are ready to be viewed.

Sincerely,

Konstantinos Kormas
Editor, Microbiology Spectrum

Journals Department
Supplemental table 1-16: Accept
Supplemental Figure: Accept